# Health effects associated with chewing tobacco: a Burden of Proof study

Gabriela F. Gil[1], Jason A. Anderson[1], Aleksandr Aravkin[1,2,3], Kayleigh Bhangdia[1], Sinclair Carr [1], Xiaochen Dai [1,2], Luisa S. Flor [1,2], Simon I. Hay [1,2], Matthew J. Malloy[1], Susan A. McLaughlin[1], Erin C. Mullany[1], Christopher J. L. Murray [1,2], Erin M. O'Connell[1], Chukwuma Okereke[1], Reed J. D. Sorensen[1], Joanna Whisnant[1], Peng Zheng[1,2] & Emmanuela Gakidou [1,2] ✉

Chewing tobacco use poses serious health risks; yet it has not received as much attention as other tobacco-related products. This study synthesizes existing evidence regarding the health impacts of chewing tobacco while accounting for various sources of uncertainty. We conducted a systematic review and meta-analysis of chewing tobacco and seven health outcomes, drawing on 103 studies published from 1970 to 2023. We use a Burden of Proof meta-analysis to generate conservative risk estimates and find weak-to-moderate evidence that tobacco chewers have an increased risk of stroke, lip and oral cavity cancer, esophageal cancer, nasopharynx cancer, other pharynx cancer, and laryngeal cancer. We additionally find insufficient evidence of an association between chewing tobacco and ischemic heart disease. Our findings highlight a need for policy makers, researchers, and communities at risk to devote greater attention to chewing tobacco by both advancing tobacco control efforts and investing in strengthening the existing evidence base.

Tobacco control efforts, including those delineated by the WHO Framework Convention on Tobacco Control (FCTC), outline the need to conduct research and implement effective policies that address a range of tobacco products[1,2]. While less ubiquitous than cigarettes, chewing tobacco use persists as a global public health challenge despite being incorporated in the FCTC statutes of 138 countries[1–3]. Chewing tobacco, a form of smokeless tobacco that is masticated by the users, encompasses a range of products, including gutkha, mainpuri, and zarda[4]. It is frequently used in combination with betel quid or other such additives[5]. As of 2019, an estimated 273.9 million individuals globally used chewing tobacco products[3]. Over 83% of chewing tobacco users resided in South Asia, including 185.8 million individuals in India and 25.7 million in Bangladesh[3]. In many of the countries with the highest usage, the use of chewing tobacco and other smokeless tobacco products is associated with important cultural practices or social norms[6,7]. The global prevalence of chewing tobacco use has

increased since 1990 in contrast to patterns of reduced smoking prevalence, and chewed tobacco products appear to be particularly popular among individuals aged 15-19 years in most countries[3]. Notably, in countries like Bangladesh with some of the highest overall rates of use, prevalence among females appears to steadily increase with age, leading to roughly 50% of females aged 80-84 chewing tobacco[3]. These trends highlight an urgent need to better integrate chewing tobacco-related considerations into existing and new tobacco control measures.

Despite widespread consensus on the harms of other tobacco products, the health risks of chewing tobacco have been less studied and are less well understood. Among some communities that have historically used chewing tobacco, smokeless tobacco is believed to aid congestion, assist with headaches, and alleviate stress[7–9]. In 2012, the 100[th] International Agency for Research on Cancer monograph outlined the association between smokeless tobacco products,

[1]Institute for Health Metrics and Evaluation, University of Washington, Seattle, WA, USA. [2]Department of Health Metrics Sciences, School of Medicine, University of Washington, Seattle, WA, USA. [3]Department of Applied Mathematics, University of Washington, Seattle, WA, USA. ✉e-mail: gakidou@uw.edu

including but not limited to chewing tobacco, and oral cancer, determining smokeless tobacco to be a carcinogen[10,11]. More recent literature regarding these health outcomes has been heterogeneous in both the results and quality of the studies[8,12,13]. Few studies focus specifically on chewing tobacco, despite substantial variation in observed risk depending on the composition of the tobacco products and mode of use. Furthermore, attempts to synthesize recent literature often do not appropriately account for between-study variation and few, if any, examine the potential of publication bias affecting their results[8,12,13]. There has also been much less attention on the relationship between chewing tobacco and cardiovascular health outcomes, like ischemic heart disease and stroke, and the evidence that does exist has many of the same limitations as the data on cancer outcomes[8,14–16].

The focus on smokeless tobacco, a broad and ambiguous grouping of tobacco products, in existing attempts to quantify health risks has also obscured the distinct risk patterns associated with chewed and non-chewed forms of smokeless tobacco. In this study, we conducted a comprehensive, systematic review and meta-analysis of the relationship between chewing tobacco specifically, distinct from other forms of smokeless tobacco, and seven health outcomes. We examined the two chewing tobacco risk-outcome pairs already included in previous iterations of the Global Burden of Diseases, Injuries, and Risk Factors Study (GBD): lip and oral cavity cancer and esophageal cancer[17]. In addition, we quantified the strength of evidence associating chewing tobacco with five additional outcomes to evaluate potentially incorporating these outcomes into the comparative risk assessment framework used by GBD to estimate the annual disease burden attributable to chewing tobacco. To the best of our knowledge, the present study marks the most up-to-date and comprehensive evaluation of the evidence underlying the health effects of chewing tobacco, specifically.

Drawing upon the Preferred Reporting Items for Systematic Reviews and Meta-Analyses (PRISMA) guidelines, we conducted three systematic reviews to capture all available peer-reviewed literature pertaining to chewing tobacco and five types of head and neck cancers, ischemic heart disease, and stroke. We searched three databases, PubMed, Global Index Medicus, and Web of Science, from January 1, 1970, through January 30, 2023, regardless of language of publication. Based on these reviews, we identified cohort and case-control studies that either reported a measure of association or contained sufficient information to calculate a measure of association between the incidence or mortality of the identified outcomes for tobacco chewers relative to individuals who do not chew tobacco. We extracted measures of associations and key study characteristics. Using the meta-regression–Bayesian, regularized, trimmed (MR-BRT) tool, we estimated a pooled relative risk for each outcome that incorporated significant sources of systematic bias, within-study correlation, and between-study heterogeneity[18]. As presented in previous Burden of Proof studies, this approach better incorporates the uncertainty evident in existing literature than traditional meta-analytic approaches, which is particularly crucial for risk factors like chewing tobacco where the existing literature is quite varied[18–23]. We then applied the Burden of Proof Risk Function (BPRF) analytic approach to evaluate the effect size and the strength of the evidence for the relative risk estimates. This approach involves deriving a conservative estimate of the minimum risk associated with chewing tobacco that is consistent with existing evidence, which is in turn translated into a star rating reflecting the evidence of association[18].

Through this approach, this work presents estimates of association between chewing tobacco and stroke, ischemic heart disease, lip and oral cavity cancer, esophageal cancer, laryngeal cancer, nasopharyngeal cancer, and other pharynx cancer and characterizes the available evidence underpinning these associations. Of the seven outcomes evaluated, six have evidence of an association with chewing tobacco and represent priority areas for policy makers, physicians, and public health advocates for improving regulation surrounding chewing tobacco marketing, taxation, cessation support, and other related measures. However, the current landscape of evidence informing the estimated relationships with chewing tobacco is weak and highlights key areas for future research, including a need for large, high-quality prospective cohort studies to strengthen our understanding of chewing tobacco's health burden. As the number of chewing tobacco users continues to increase in many low- and middle-income countries, our findings reflect the importance of better incorporating chewing tobacco into broader tobacco control efforts that have been instrumental in reducing the burden of smoked tobacco (Table 1).

## Results

### Overview

In this systematic review and meta-analysis, we evaluated the strength of the evidence and association between seven health outcomes and the use of chewing tobacco products. In total, we evaluated 4,340 records identified in PubMed, Web of Science, and Global Index Medicus. An additional 251 records were identified for screening through citation searching of other identified records. We excluded 4,480 records from our analysis according to our predetermined exclusion criteria (Supplementary Information 1.2). The remaining 111 publications, which report on 103 studies, use prospective cohort, retrospective cohort, or case-control study designs and provide data on the association between the outcome of interest and chewing tobacco broadly, or specific chewed forms of tobacco, among a general population. We had 81.5%, 92.9%, and 84.6% reviewer concordance during title/abstract screening and 88.9%, 94.4%, and 97.7% reviewer concordance for full-text screening of studies on head and neck cancers, stroke, and ischemic heart disease, respectively. Details on each study included—i.e., study design, number of participants, exposure and outcome definitions, and confounders adjusted for—can be found in the Supplementary Information. PRISMA diagrams for each of the systematic reviews are presented in Supplementary Figures S1-S3.

### Stroke

For stroke, our systematic review identified three unique case-control studies with four eligible observations from India and Bangladesh (Fig. 1; Supplementary Table S4)[24–26]. These studies included sample sizes ranging from 430,596 to 160. One of these observations was reported for chewing tobacco broadly, while the other three more narrowly defined chewing tobacco as betel quid with tobacco, gutkha, or pan with tobacco, respectively. Other study characteristics are described in the Supplementary Information. Because fewer than 10 relevant observations were included, this model was not eligible for trimming potential outliers since the data sparsity made outliers difficult to detect.

The BPRF analysis suggests that using chewing tobacco increases an individual's risk of stroke by a conservative minimum of 16% (BPRF = 1.16) as a two-star risk-outcome pair (Table 2). In other words, we found that the existing evidence suggesting a harmful relationship between chewing tobacco use and the risk of stroke is rated as weak. Despite the limited data available, the observations included were consistent ($\gamma = 4.7 \times 10^{-6}$; Supplementary Table S12) in their findings of a harmful association. The estimated relative risk incorporating between-study heterogeneity is 1.46 (95% uncertainty interval (UI) = 1.11–1.93; Table 2; Fig. 2). The covariate selection algorithm did not find any of the candidate covariates significant (Table 2). The results were robust to various sensitivity analyses, although the sparsity of observations limited the sensitivity analyses that were feasible (Fig. 3; Supplementary Table S13). We did not detect publication bias in the primary analysis or in any of the sensitivity analyses (Table 2; Supplementary Table S1).

## Table 1 | Policy Summary

| | |
|---|---|
| Background | Chewing tobacco has historically received less public health attention than its smoked counterparts despite its inclusion in the WHO Framework Convention for Tobacco Control and high rates of use in some regions and communities where it is associated with important cultural practices. Chewing tobacco is often categorized under the vast umbrella of smokeless tobacco. Previous meta-analyses have found that studies on the health risks related to the broader category of smokeless tobacco conducted in south Asian countries—where chewing tobacco products dominate—typically report greater associated health risks. The present meta-analysis synthesizes the available evidence on chewing tobacco use and its association with seven health outcomes. We applied a meta-regression framework to data extracted through three comprehensive systematic reviews, spanning 53 years of peer-reviewed literature indexed by three major databases. Using Burden of Proof methodology, we generated conservative estimates of health risks associated with chewing tobacco use consistent with existing evidence—rigorously quantifying and incorporating measures of between-study heterogeneity, accounting for potential within-study correlation, and testing and adjusting for potential systematic bias. |
| Main findings and limitations | Our conservative interpretation of available data finds weak-to-moderate evidence of harmful associations between chewing tobacco use and esophageal cancer and stroke. The risk of these outcomes is at least 2-16% higher among tobacco chewers than non-chewers. However, we also detected a large degree of between-study heterogeneity in the evidence underpinning chewing tobacco's association with esophageal cancer. High estimates of between-study heterogeneity were also observed in the weak existing evidence suggesting a relationship between chewing tobacco and lip and oral cavity cancer, laryngeal cancer, nasopharynx cancer, and other pharynx cancer. Ischemic heart disease has insufficient evidence of a significant risk-outcome relationship with chewing tobacco. There is a need for large high-quality prospective cohort studies to further our understanding of the health burden of chewing tobacco, particularly for risk-outcome pairs with limited or inconsistent evidence. Some limitations of our approach include the variability of exposure and outcome definitions. We included studies that reported on any chewing tobacco products because local chewing tobacco products vary depending on study location. However, the composition of different local chewing tobacco products may impact their health effects. We also did not restrict the use of aggregate outcome definitions in input data for the head and neck cancer outcomes analyzed because of the frequency with which cancer sub-types were grouped together. We evaluated the risk of systematic bias introduced by these two limitations. More broadly, we evaluated the health risks associated with chewing tobacco compared to not chewing tobacco without considering dosage. Accordingly, the dose-response relationship of these risk-outcome associations is an important area for future work. |
| Policy implications | In contrast to global trends of reduced smoking prevalence, the use of chewing tobacco has increased in recent decades, especially among youth and older women in some areas of the world. Despite the relative paucity of data, our analysis indicates that chewing tobacco use may increase the risk of stroke, esophageal cancer, lip and oral cavity cancer, nasopharynx cancer, other pharynx cancer, and laryngeal cancer. Our research further highlights the need for more large prospective cohort studies on the risks associated with chewing tobacco to bolster our understanding of its potential health consequences. These findings highlight the urgent need to better incorporate chewing tobacco into new and existing tobacco control efforts and to expand research efforts investigating the health burden of chewing tobacco. |

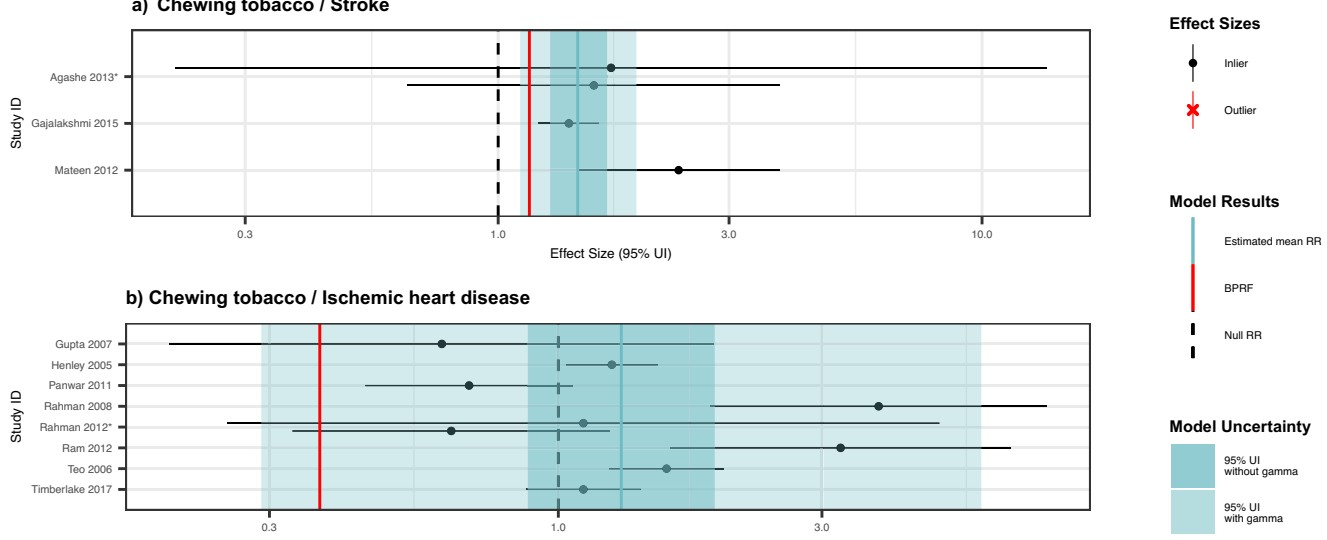

**Fig. 1 | Forest plots of underlying data for chewing tobacco and two cardio-vascular outcomes.** These forest plots depict the estimated mean relative risk (blue vertical line) and its 95% uncertainty interval (blue shaded intervals) for the association between chewing tobacco and stroke (panel **a**) and for the association between chewing tobacco and ischemic heart disease (panel **b**) and the underlying data points. The narrower darker blue intervals correspond to the 95% uncertainty interval estimated without accounting for between-study heterogeneity in accordance with traditional meta-analytic approaches. The light blue intervals correspond to the 95% uncertainty interval that incorporates between-study heterogeneity and the uncertainty around it. Similarly, the red vertical lines are the Burden of Proof Risk Function (BPRF), which correspond to the 5th quantile and is

used to derive our risk-outcome score (ROS) for risk-outcome pairs in which the darker blue intervals (the 95% uncertainty interval without between-study heterogeneity) do not include the null value at relative risk = 1. The black dotted vertical lines reflect the null relative risk at 1. The black data points and horizontal lines each correspond to an effect size and 95% uncertainty interval from the study noted in on the y-axes that were included in the models. Neither model qualified for trimming, so no observations are marked with red Xs. Studies noted with an asterisk include effect sizes from overlapping samples whose uncertainty interval was scaled based on the number of overlapping observations to avoid overrepresenting one sample in the models.

**Table 2 | Strength of the evidence for the relationship between chewing tobacco and the seven health outcomes analyzed**

| Health outcome | RR (95% UI without γ) | RR (95% UI with γ) | BPRF | ROS | Star rating | Pub. bias | No. of studies | Selected bias covariates | Pair in GBD |
|---|---|---|---|---|---|---|---|---|---|
| Stroke | 1.46 (1.28, 1.68) | 1.46 (1.11, 1.93) | 1.16 | 0.07 | ⭐⭐ | No | 3 | None | N |
| Esophageal cancer | 2.14 (1.77, 2.57) | 2.14 (0.89, 5.15) | 1.02 | 0.01 | ⭐⭐ | No | 22 | Maximally adjusted; Adjusted for smoking, age, and sex | Y |
| Lip and oral cavity cancer | 3.64 (3.00, 4.41) | 3.64 (0.66, 19.95) | 0.87 | -0.07 | ⭐ | No | 70 | Chewing tobacco product; Study subpopulation | Y |
| Larynx cancer | 2.66 (1.98, 3.57) | 2.66 (0.52, 13.63) | 0.68 | -0.20 | ⭐ | No | 24 | Aggregate outcome definition; Adjusted for age and sex | N |
| Nasopharynx cancer | 2.50 (1.79, 3.49) | 2.50 (0.49, 12.66) | 0.64 | -0.22 | ⭐ | No | 17 | Maximally adjusted; Adjusted for age and sex | N |
| Other pharynx cancer | 2.33 (1.80, 3.01) | 2.33 (0.45, 12.04) | 0.59 | -0.27 | ⭐ | No | 31 | Aggregate outcome definition; Adjusted for age and sex | N |
| Ischemic heart disease | 1.30 (0.88, 1.92) | 1.30 (0.29, 5.83) | N/A | N/A | | No | 8 | None | N |

The reported relative risk (RR) and its 95% uncertainty interval (UI) reflect the risk an individual who uses chewing tobacco has of developing the outcome of interest relative to that of someone who does not use chewing tobacco. Gamma (γ) quantifies the estimated between-study heterogeneity of included observations. We report two separate 95% UIs, one that is estimated without incorporating between-study heterogeneity (γ) and one that does account for this source of uncertainty—"95% UI with γ." The Burden of Proof Risk Function (BPRF) is calculated for risk-outcome pairs that were found to have significant relationships at an 0.05 level of significance when between-study heterogeneity is not incorporated. The BPRF corresponds to the 5th quantile estimate of relative risk accounting for between-study heterogeneity closest to the null for each risk–outcome pair, and it reflects the most conservative estimate of excess risk associated with chewing tobacco that is consistent with the available data. Since we define chewing tobacco exposure as a dichotomous risk factor, i.e., an individual either currently chews tobacco or does not, the risk-outcome score (ROS) is calculated as the signed value of natural log(BPRF) divided by two. Negative ROSs indicate that the evidence of the association is very weak and inconsistent. For ease of interpretation, we have transformed the ROS and BPRF into a star rating (1–5) with a higher rating representing a larger effect with stronger evidence. A zero-star rating is assigned to risk-outcome pairs whose RR 95% uncertainty interval without consideration of between-study heterogeneity crosses 1. The potential existence of publication bias, which, if present, would affect the validity of the results, was tested using Egger's Regression. Included studies represent all available relevant data identified through our systematic reviews from January 1970 through January 2023. The selected bias covariates were chosen for inclusion in the model using an algorithm that systematically detects bias covariates that correspond to significant sources of bias in the observations included. If selected, the observations were adjusted to better reflect the gold standard values of the covariate. See the Supplementary Information for more information about the candidate bias covariates.

## Ischemic heart disease

For ischemic heart disease, we identified eight unique studies with nine eligible observations used to inform our primary analysis (Fig. 1; Supplementary Table S4)[27–35]. Seven of the studies were conducted in the United States, Bangladesh, and India, while the eighth study drew upon data from 52 locations. These studies included two prospective cohort studies and six case-control studies. We found 11.1% (1/9) of the eligible observations solely included men and none were female-specific. Of the eligible observations, 66.7% (6/9) reported effect sizes for chewing tobacco broadly, and 33.3% (3/9) specifically reported on the effects of current use. Other study characteristics are described in the Supplementary Information.

We did not find sufficient evidence of a significant association between chewing tobacco use and the risk of ischemic heart disease based on our conservative interpretation of the data that yielded a zero-star risk-outcome association (Table 2). Risk-outcome pairs are not eligible for potential inclusion in the GBD if the conventional estimate of relative risk, in which the uncertainty is estimated without considering between-study heterogeneity, does not reflect a statistically significant relationship. This is the case with chewing tobacco and ischemic heart disease, which our analysis found to have an estimated relative risk of 1.30 (0.29–5.83) with between-study heterogeneity and 1.30 (0.88–1.92) without between-study heterogeneity (Table 2; Figs. 1 and 2). No covariates were selected as significant by the bias covariate algorithm, and no publication bias was detected (Table 2). Since there were fewer than 10 included observations, we did not implement 10% trimming. These results were robust to various sensitivity analyses, including restricting data point inclusion based on exposure definitions and sample sizes (Fig. 3; Supplementary Table S14).

## Esophageal cancer

We identified 22 unique studies with 31 eligible observations from three locations that were used to inform our primary analysis of the association between esophageal cancer and chewing tobacco (Fig. 4; Supplementary Table S4)[36–59]. These studies included one prospective cohort, one case-cohort, and 20 case-control studies with sample sizes ranging from 219,444 to 99. We found that 29.0% (9/31) of the eligible

observations were specific to male participants, while 12.9% (4/31) were specific to female participants, and the remainder were reported for both male and female participants. A total of 54.8% (17/31) of the observations reported effect sizes for chewing tobacco broadly, with the remainder defining their exposure as the use of a specific chewing tobacco product, including betel quid with tobacco (5/31). Out of the 31 eligible observations, 16.1% (5/31) used an aggregate outcome definition that included other head and neck cancers and were flagged by a bias covariate (Supplementary Table S9). Other study characteristics are described in the Supplementary Information.

Based on these data, the BPRF analysis suggests that the use of chewing tobacco increases the risk of esophageal cancer by at least 2% with a BPRF of 1.02 (Table 2). Further, the relationship between esophageal cancer and chewing tobacco use is categorized as a two-star risk-outcome pairing (Table 2)[18]. When the heterogeneity between studies is not considered (reflecting a traditional meta-analytic approach), chewing tobacco users were found to have a 2.14 (1.77–2.57)-fold increased risk of esophageal cancer (Table 2; Fig. 5). Some between-study heterogeneity was observed (γ = 0.092; Supplementary Table S12), which resulted in a larger 95% UI (0.89–5.15) when incorporated into the estimate of uncertainty surrounding the relative risk (Table 2; Fig. 5). Moreover, the covariate selection algorithm identified a significant difference between data points that were maximally controlled and those that were not and a further difference between data points that controlled for age, sex, and smoking, and those that did not control for smoking, so the two corresponding covariates were used to accordingly adjust the observations included (Table 2).

The ROS, BPRF, and resulting star rating were robust to the removal of all potential covariates (Fig. 3; Supplementary Table S15). When the analysis was limited to only the 12 observations that compare current chewers to a reference group of non-chewers or never chewers, which most closely aligns with the canonical chewing tobacco definition, the between-study heterogeneity decreased substantially (γ < 0.001), and the relationship became a three-star association (ROS = 0.25). Other changes in our model parameters, standard error adjustments, and data point inclusion resulted in one-star associations (Fig. 3; Supplementary Table S15). We did not detect publication bias in

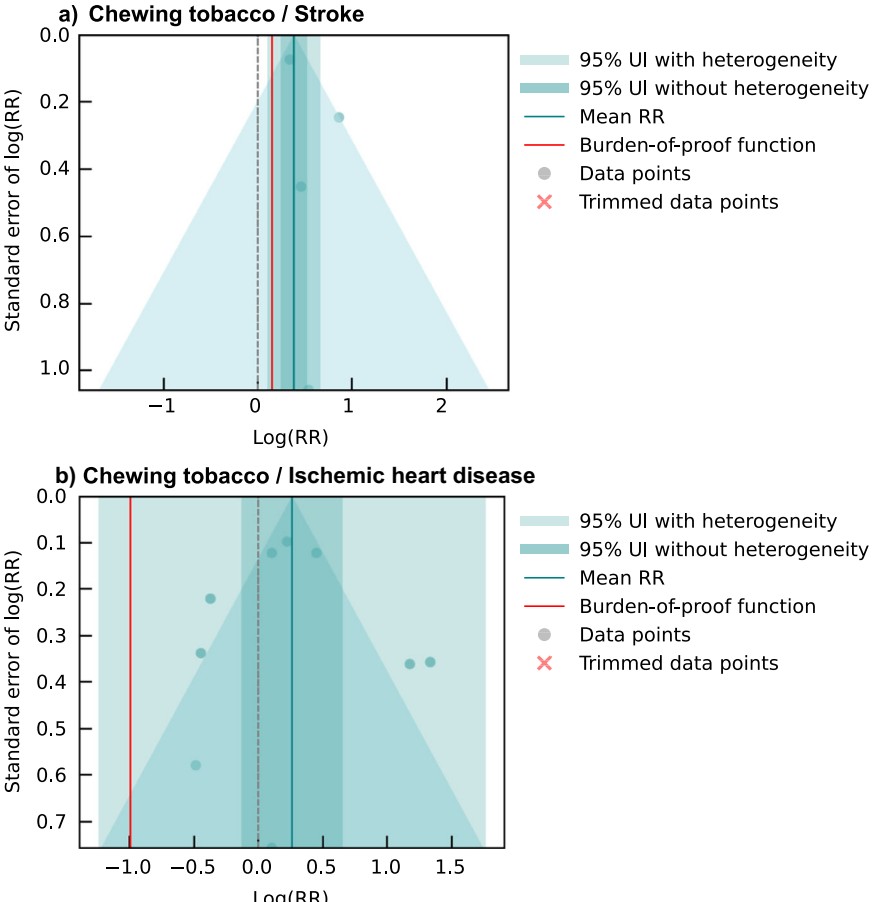

**Fig. 2 | Modified funnel plots for chewing tobacco and two cardiovascular disease outcomes.** These modified funnel plots show the residuals of the reported mean relative risk (RR) relative to 0, the null value, on the x-axis and the residuals of the standard error, as estimated from both the reported standard error and gamma, relative to 0 on the y-axis for the association between chewing tobacco and stroke (panel **a**) and between chewing tobacco and ischemic heart disease (panel **b**). The light blue vertical interval corresponds to the 95% uncertainty interval incorporating between-study heterogeneity; the dark blue vertical interval corresponds to the 95% uncertainty interval without between-study heterogeneity; the dots are each included observation; the red Xs are outliered observations if relevant; the grey dotted line reflects the null log(RR); the blue line is the mean log(RR) for chewing tobacco and the outcome of interest; the red line is the burden of proof function at the 5[th] quantile for these harmful risk-outcome associations.

the esophageal cancer data used in the primary analysis when trimming 10% of observations (Table 2; Fig. 5) nor in any of the sensitivity analyses, regardless of the use of trimming (Supplementary Table S15).

## Lip and oral cavity cancer

For lip and oral cavity cancer, our head and neck cancer systematic review identified 70 unique studies with 106 eligible observations from 10 locations (Fig. 6; Supplementary Table S4)[36,37,41,44,47,50,51,53,60–125]. These studies included two prospective cohort, one case-cohort, one nested case-control, and 66 case-control studies with sample sizes ranging from 219,444 to 76. We found that 72.6% (77/106) of the observations included both male and female participants and 57.6% (61/106) of the observations reported effect sizes for chewing tobacco broadly. There was substantial variation in the specific chewing tobacco products examined by the other 45 observations, including 12 betel quid with tobacco and 12 gutkha (Supplementary Table S4). Out of the 106 eligible observations included in the lip and oral cavity cancer model, 25.5% (27/106) used an aggregate outcome definition. Other study characteristics are described in the Supplementary Information.

With an estimated ROS of -0.07, the relationship between lip and oral cavity cancer and chewing tobacco use reflects a one-star risk-outcome association (Table 2; Fig. 5). This conservative interpretation of the evidence, which incorporates several sources of uncertainty, results in a relative risk of 3.64 (0.66–19.95; Table 2). The large

uncertainty interval reflects a considerable degree of observed between-study heterogeneity (γ = 0.53; Supplementary Table S12), potentially due to variations in exposure and outcome definitions used across the input studies (Supplementary Table S4). With a conventional uncertainty interval, which does not fully incorporate between-study heterogeneity, the estimated relative risk and associated uncertainty is 3.64 (3.00–4.41; Table 2). The covariate selection algorithm flagged observations that were for specific chewing tobacco products, compared to those for chewing tobacco broadly, and observations that were derived from study subpopulations for adjustment in the model (Table 2). The results were robust to changes in model parameters, data adjustments, and data point inclusion (Fig. 3; Supplementary Table S16). Among only studies that limited their samples to non-smokers and those studies conducted in Asian countries, there was an increase to a two-star rating for the relationship between chewing tobacco and lip and oral cavity cancer (Fig. 3; Supplementary Table S16). We did not detect publication bias after trimming 10% of observations or in any of the sensitivity analyses that did not involve trimming.

## Laryngeal cancer

For laryngeal cancer, we identified 24 unique studies with 30 eligible observations from three locations (Fig. 7; Supplementary Table S4)[37,44,47,51,53,61,63,65,66,71–74,101,105,106,109,113,118,120,126–129]. These studies

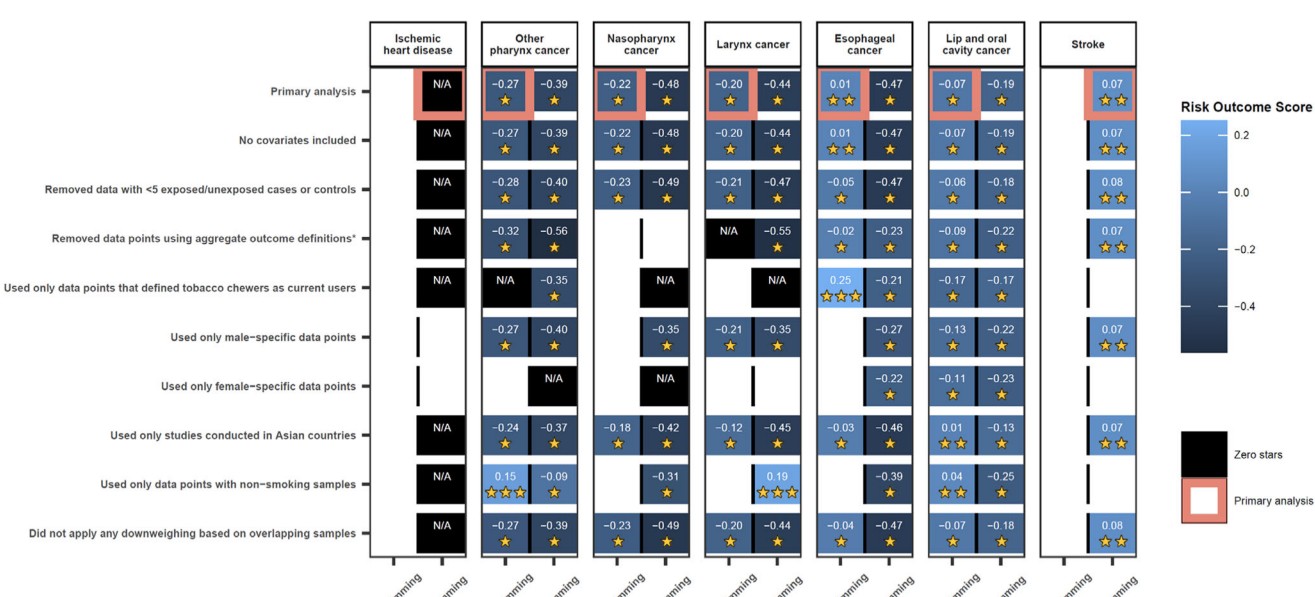

**Fig. 3 | Summarized results of various sensitivity analyses conducted across all seven health outcomes.** This heatmap reports the results of the various sensitivity analyses conducted for the seven health outcomes. The details of each, beyond the description on the y-axis, are described in detail in the Supplementary Information. Each model parameter or change in data inclusion was tested both incorporating 10% trimming and with no trimming, as depicted along the x-axis. It was only feasible to test models with more than three observations, and 10% trimming could only be implemented for models with more than 10 observations. The model combinations that were not possible to test are depicted as white boxes. The color of the blue boxes and number in each box corresponds to the risk-outcome score (ROS) calculated for models in which the estimates of association without incorporating between-study heterogeneity were statistically significant. Black boxes depict models that did not pass this threshold and, thus, ROS did not apply (N/A).

For models that did pass this threshold, the ROS reflects a conservative interpretation of the data that aligns with the Burden of Proof approach incorporating between-study heterogeneity and other sources of uncertainty. The ROS translates into a star rating from 1 to 5 stars. The star rating for each model result is reported as the yellow stars in each box. A one-star association suggests that there is weak evidence supporting estimates of an association between the risk and outcome. A two-star association reflects that there is weak-to-moderate evidence suggesting an association between the risk and outcome, and additional stars illustrate increasing strength of evidence. The pink outlined boxes highlight our primary models with the trimming approach that corresponds to the number of observations (10% trimming for models with more than 10 observations; no trimming for models with fewer observations).

included one prospective cohort study and 23 case-control studies with sample sizes ranging from 219,444 to 215. A total of 46.7% (14/30) of the eligible observations were derived from male-only study samples. In addition, 73.3% (22/30) of the observations reported effect sizes for chewing tobacco broadly, with betel quid with tobacco (n = 3) and gutkha (n = 2) also represented. The candidate bias covariate for aggregate outcome definitions, including other head and neck cancers flagged 60.0% (18/30) of the observations (Supplementary Table S9). Other study characteristics are described in the Supplementary Information.

We found weak evidence of a relationship between chewing tobacco use and the risk of laryngeal cancer based on our conservative interpretation of the data (ROS = -0.20; Table 2; Fig. 5). Any risk-outcome pair with an ROS less than 0 with a significant association in the traditional fixed effects model without between-study heterogeneity is categorized as having a one-star association, in which the evidence of a relationship between the risk factor and the outcome is rated as weak when accounting for various forms of uncertainty[18]. Thus, our analysis of the relationship between chewing tobacco and laryngeal cancer yielded a one-star risk-outcome association. The estimated relative risk was 2.66 (0.52–13.63) when accounting for various sources of bias and uncertainty (Table 2). Observations that were generated for aggregate outcome definitions and those that did not control for age and sex were adjusted for in the model, as these covariates were found to be significant (Table 2). When limiting the study to only the 8 observations derived from samples of non-smokers and not applying the 10% trimming algorithm, the association between

chewing tobacco and laryngeal cancer was estimated to be a three-star risk-outcome pair. The overall one-star relationship remained consistent across various other sensitivity analyses, including those that entailed no covariates, while restricting included observations to only laryngeal cancer-specific outcome definitions and restricting included observations to only current chewers both resulted in zero-star relationships (Fig. 3; Supplementary Table S17). No publication bias was detected with 10% trimming (Fig. 5).

## Nasopharyngeal cancer
We identified 17 unique studies with 24 eligible observations from three locations that are used to estimate the relationship between chewing tobacco and nasopharynx cancer (Fig. 7; Supplementary Table S4)[44,47,51,53,61,63,65,66,71,72,74,105,112,113,118,120,123,124,130]. All of the studies used case-control study designs and had sample sizes ranging from 219,444 to 141. Only two of the 24 observations that included nasopharyngeal cancer cases were specific to this outcome. Other study characteristics are described in the Supplementary Information.

Using the BPRF approach, we found weak evidence of a significant relationship between chewing tobacco use and the risk of nasopharyngeal cancer (ROS = -0.22; Table 2; Fig. 5). The estimated relative risk inclusive of between-study heterogeneity of this one-star risk-outcome pair is 2.50 (0.49–12.66; Table 2), due in part to substantial heterogeneity (γ = 0.34; Supplementary Table S12). The covariate selection algorithm identified significant differences in data that were not maximally controlled (compared to maximally controlled) and data that were not controlled for age and sex, so these cascading covariates

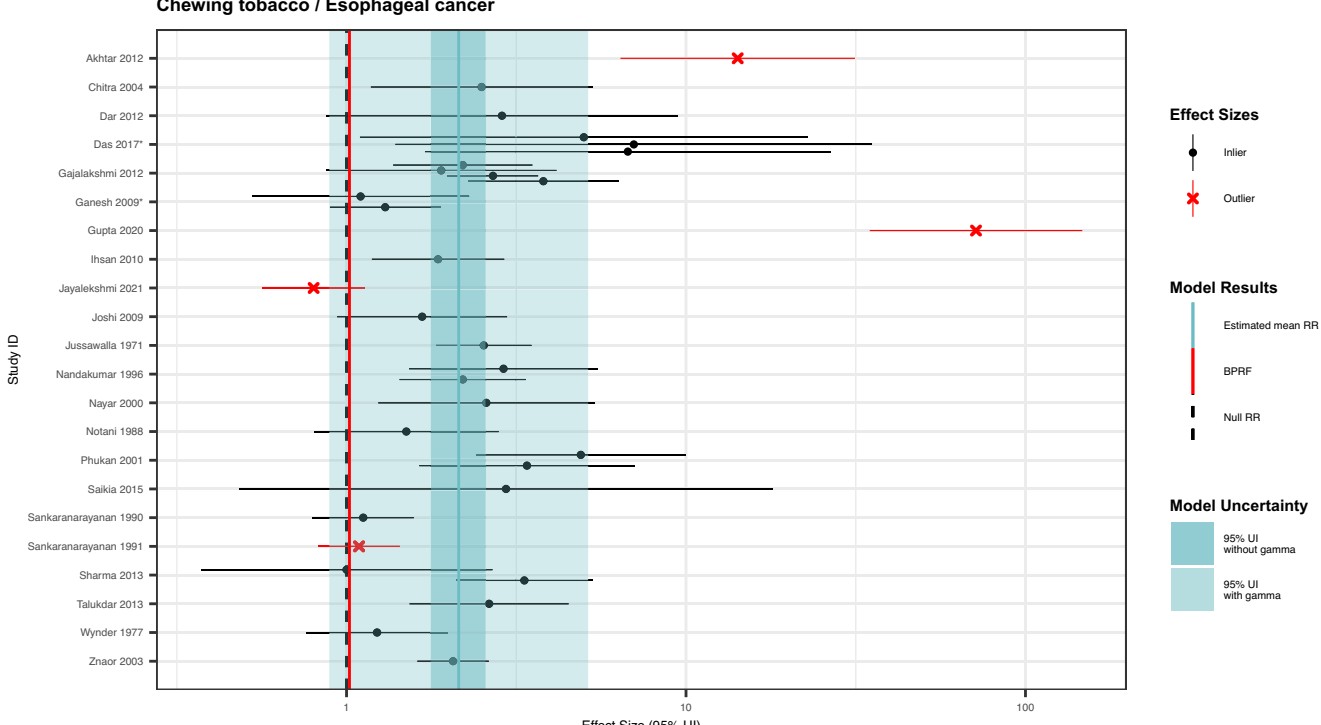

**Fig. 4 | Forest plot of underlying data for chewing tobacco and esophageal cancer.** This forest plot depicts the estimated mean relative risk (blue vertical line) and its 95% uncertainty interval (blue shaded intervals) for the association between chewing tobacco and esophageal cancer and the data points used to produce our primary results. The narrower darker blue interval corresponds to the 95% uncertainty interval estimated without accounting for between-study heterogeneity in accordance with traditional meta-analytic approaches. The light blue interval corresponds to the 95% uncertainty interval that incorporates between-study heterogeneity and the uncertainty around it in alignment with our Burden of Proof meta-analytic approach. Similarly, the red vertical line is the Burden of Proof Risk Function (BPRF), which corresponds to the 5th quantile and is the estimate from which our risk-outcome score (ROS) is derived for risk-outcome pairs in which the darker blue interval (the 95% uncertainty interval without between-study heterogeneity) does not include the null value at relative risk = 1. The black dotted vertical line reflects the null relative risk at 1. The black data points and horizontal lines each correspond to an effect size and 95% uncertainty interval from the study noted in on the y-axis that was included in the model. The red Xs and horizontal lines correspond to effect sizes and 95% uncertainty intervals from the studies on the y-axis that were automatically trimmed by the trimming algorithm based on deviation from the mean. Studies noted with an asterisk include effect sizes from overlapping samples whose uncertainty interval was scaled based on the number of overlapping observations to avoid overrepresenting one sample in the model.

were included in the model (Table 2). No publication bias was detected, and the one-star relationship remained consistent across various sensitivity analyses, barring limiting the included observations to only current chewers or female-specific observations in which estimated relative risk without between-study heterogeneity crossed the null. (Figs. 3 and 5; Supplementary Table S18).

## Other pharynx cancer
We identified 31 unique studies with 43 eligible observations from four locations (Fig. 7; Supplementary Table S4)[36,44,47,51,53,61,63,65,66,71–74,84,89,95,97,103–106,112,113,115,118–120,123,124,126,127,129,131]. These studies included one prospective cohort study and 30 case-control studies with sample sizes ranging from 219,444 to 123. Out of 43 observations, 12 were specific to other pharynx cancer. Other study characteristics are described in the Supplementary Information.

We found weak evidence of a relationship between chewing tobacco use and the risk of other pharynx cancer (ROS = -0.27; Table 2; Fig. 5). As a one-star risk-outcome pair, there was substantial between-study heterogeneity (γ = 0.43; Supplementary Table S12) and an estimated relative risk with between-study heterogeneity of 2.33 (0.45–12.04; Table 2). A sensitivity analysis conducted by limiting the included observations to only those from non-smoking study samples substantially lowered the observed between-study heterogeneity (γ = 0.20). An analysis with this data subset and 10% trimming found stronger evidence of a relationship between chewing tobacco

use and other pharynx cancer among non-smokers as a three-star risk-outcome pair and a relative risk with between-study heterogeneity of 4.38 (1.09–17.58; Supplementary Table S19) The observations that used aggregate outcome definitions and those that were not controlled for age and sex were accounted for through included covariates selected by the covariate selection algorithm (Table 2). We trimmed 10% of data as outliers and did not detect publication bias (Fig. 5). The one-star relationship was robust to most other sensitivity analyses, except for using only female-specific observations and using only current users as the exposed group with 10% trimming (Fig. 3; Supplementary Table S19).

## Discussion
This study systematically synthesizes available evidence regarding the health risks associated with chewing tobacco, distinct from other forms of smokeless tobacco, and evaluates the consistency of said evidence. Across all seven health outcomes, six health outcomes—esophageal cancer, lip and oral cavity cancer, laryngeal cancer, nasopharynx cancer, other pharynx cancer, and stroke—were found to have weak, albeit sufficient, evidence supporting an association with chewing tobacco. Laryngeal cancer, nasopharynx cancer, lip and oral cavity cancer, and other pharynx cancer were found to have one-star risk-outcome associations with chewing tobacco use. Esophageal cancer and stroke were found to be two-star risk-outcome pairs, whereupon our conservative interpretation of the available data indicated that chewing tobacco increases the risk of these outcomes by at

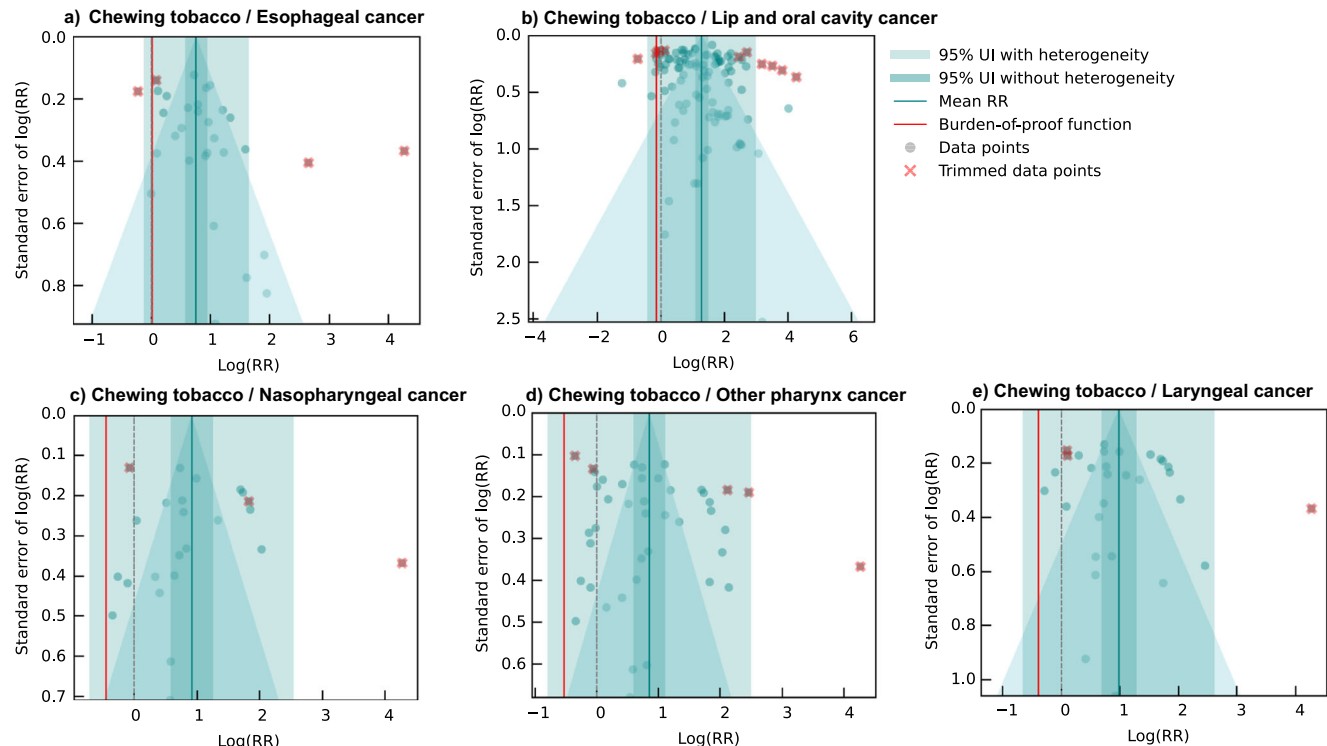

**Fig. 5 | Modified funnel plots for chewing tobacco and five head and neck cancer outcomes.** These modified funnel plots show the residuals of the reported mean relative risk (RR) relative to 0, the null value, on the x-axis and the residuals of the standard deviation, as estimated from both the reported standard deviation and gamma, relative to 0 on the y-axis. Each funnel plot corresponds to a different model for esophageal cancer (panel **a**), lip and oral cavity cancer (panel **b**), nasopharyngeal cancer (panel **c**), other pharynx cancer (panel **d**), and laryngeal cancer (panel **e**) and their corresponding association with chewing tobacco. The light blue vertical interval corresponds to the 95% uncertainty interval incorporating between-study heterogeneity; the dark blue vertical interval corresponds to the 95% uncertainty interval without between-study heterogeneity; the dots are each included observation; the red Xs are outliered observations; the grey dotted line reflects the null log(RR); the blue line is the mean log(RR) for chewing tobacco and the outcome of interest; the red line is the burden of proof function at the 5th quantile for these harmful risk-outcome associations.

least 2-16% as derived from their estimated BPRF. The associations we found are consistent with prior results suggesting relationships between smokeless tobacco broadly and head and neck cancers and stroke based on studies in South Asia and Southeast Asia, where chewing tobacco is the dominant form of smokeless tobacco[8,12–15]. By focusing solely on chewing tobacco products, our approach reduces the impact of regional variation in smokeless tobacco use practices on our global analysis and serves to affirm the harmful effects of chewed tobacco products, specifically.

Notably, while the evidence of a harmful association between stroke and chewing tobacco appeared consistent and robust to various sensitivity analyses, the evidence supporting a harmful association between chewing tobacco use and esophageal cancer, lip and oral cavity cancer, laryngeal cancer, nasopharynx cancer and other pharynx cancer showed greater uncertainty. We identified a large degree of between-study heterogeneity for these five outcomes. This heterogeneity resulted in a high degree of sensitivity to model parameters and data point inclusion and more uncertainty in our relative risk estimates than in previous studies. Our analysis advances existing literature by better incorporating between-study variation in our derived estimates, which is particularly important for examining associations such as these where study characteristics diverge substantially and there is a paucity of gold-standard evidence[23]. Similarly, while there is consensus that chewing tobacco is a known carcinogen, existing literature focuses on its association with esophageal cancer and lip and oral cavity cancer, and its relationship to these other head and neck cancers is both less studied and less consistently demonstrated in the case-control studies that are available[8,13]. Taken together, our results underscore the need for high-quality prospective cohort studies with

greater consistency in case definitions to bolster the strength of the evidence underlying our understanding of chewing tobacco's health impacts.

Based on the present results, stroke, nasopharyngeal cancer, laryngeal cancer, and other pharynx cancer are now found to have sufficient evidence to merit consideration for inclusion in future GBD cycles. In contrast, ischemic heart disease was found to have insufficient evidence to support an association, even without taking into consideration between-study heterogeneity. These findings reflect the mixed results identified in previous smaller systematic reviews pertaining to chewing tobacco and cardiovascular diseases. Hajat et al., for example, identified one high-quality meta-analysis that reported an increased risk of ischemic heart disease among smokeless tobacco users in Asia, where chewing tobacco makes up a large portion of smokeless tobacco use, while other studies conducted in the region reported no association[8]. Another review of cardiovascular outcomes and smokeless tobacco also identified variable results for ischemic heart disease in this region, despite consistent results of an association with stroke, akin to our own findings for chewing tobacco[14].

Beyond highlighting an important research priority for academics and research funders, the BPRF, ROS, relative risk estimate, and star rating paint a comprehensive picture of the current state of evidence on the association between chewing tobacco and the seven selected health outcomes. The one- and two-star risk-outcome pairs affirm that chewing tobacco is a harmful risk factor for health outcomes of great public health significance. Physicians, public health practitioners, and tobacco control advocates can draw upon these findings to better counsel patients and advocate for better integration of chewing-tobacco-specific considerations in tobacco control policies[132].

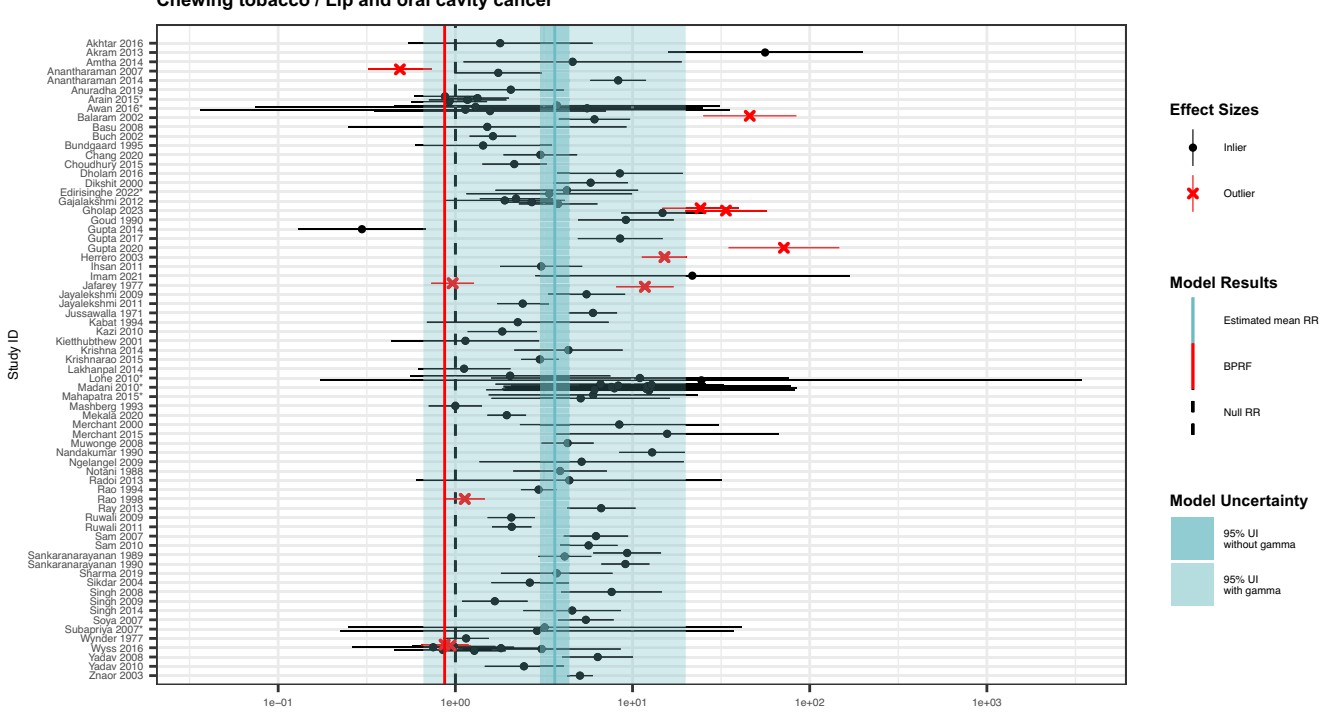

**Fig. 6 | Forest plot of underlying data for chewing tobacco and lip and oral cavity cancer.** This forest plot depicts the estimated mean relative risk (blue vertical line) and its 95% uncertainty interval (blue shaded intervals) for the association between chewing tobacco and lip and oral cavity cancer and the data points used to produce our primary results. The narrower darker blue interval corresponds to the 95% uncertainty interval estimated without accounting for between-study heterogeneity in accordance with traditional meta-analytic approaches. The light blue interval corresponds to the 95% uncertainty interval that incorporates between-study heterogeneity and the uncertainty around it in alignment with our Burden of Proof meta-analytic approach. Similarly, the red vertical line is the Burden of Proof Risk Function (BPRF), which corresponds to the 5th quantile and is the estimate

from which our risk-outcome score (ROS) is derived for risk-outcome pairs in which the darker blue interval (the 95% uncertainty interval without between-study heterogeneity) does not include the null value at relative risk = 1. The black dotted vertical line reflects the null relative risk at 1. The black data points and horizontal lines each correspond to an effect size and 95% uncertainty interval from the study noted in on the y-axis that was included in the model. The red Xs and horizontal lines correspond to effect sizes and 95% uncertainty intervals from the studies on the y-axis that were automatically trimmed by the trimming algorithm based on deviation from the mean. Studies noted with an asterisk include effect sizes from overlapping samples whose uncertainty interval was scaled based on the number of overlapping observations to avoid overrepresenting one sample in the model.

Furthermore, our evaluation of existing literature may serve to directly inform public education campaigns in communities that commonly use chewing tobacco to increase awareness of the associated harms[2]. Finally, the relative risk estimates can be used to more accurately quantify the population-level disease burden attributable to chewing tobacco, both as a whole and for each of these health outcomes[3,17]. Previous iterations of the Global Burden of Diseases, Injuries, and Risk Factors Study (GBD) drew upon six studies to estimate the relative risk used to inform the burden attributable to chewing tobacco for its two included risk-outcome pairs: lip and oral cavity cancer and esophageal cancer. These estimates of risk are now informed by 70 and 22 studies, respectively, which continue to confirm the harmful association between chewing tobacco and these outcomes for future iterations of GBD. Even with the greater compilation of data, our updated relative risk estimates are consistent with those previously used in GBD for chewing tobacco and lip and oral cavity cancer and for chewing tobacco and esophageal cancer[17]. In prior GBD rounds, sex-specific relative risks were calculated for lip and oral cavity cancer given that females were found to have higher associated risk, a pattern that was found to be present but not significant through our sex-specific sensitivity analyses (Supplementary Table S16)[17]. Of the five potential new risk-outcome pairs we evaluated, four were found to now be eligible for further consideration, which will better capture the full breadth of disease burden attributable to chewing tobacco.

Our findings are subject to a number of limitations primarily associated with the availability of data. First, our results draw upon

data that rely on a wide range of exposure definitions and settings (Supplementary Table S4). In other studies examining smokeless tobacco broadly, regional differences in risk profiles have been attributed to differences in the composition of smokeless tobacco products[8,13]. Although we limited our analysis to chewing tobacco products, these products still reflect a very heterogenous, albeit smaller, subset of the broad smokeless tobacco grouping that is more commonly used. The composition of different local chewed tobacco products may affect each product's unique risk profile, but we were not able to account for these differences due to a limited number of product-specific data points. Furthermore, some of these local products include tobacco mixed with other known independent carcinogens, such as betel nut/areca nut[5,133]. We only included data on such products if the authors explicitly noted that the exposure definition used was the product mixed with tobacco for chewing, but the use of both products together may compound health risks. The differences in risk profile may also be evident across different occupational settings, and while the breadth of existing literature is insufficient to evaluate differences in risk by occupational setting, this dimension is important to consider in future research. Furthermore, in light of unknown within-study covariance for different effect sizes reported by the same study, including for different sub-types of chewing tobacco, we elected to apply a very conservative approximation of the covariance matrix, which may present an area for future methodological development. An additional limitation is our use of a dichotomous risk definition. There is some very limited evidence to suggest that smokeless

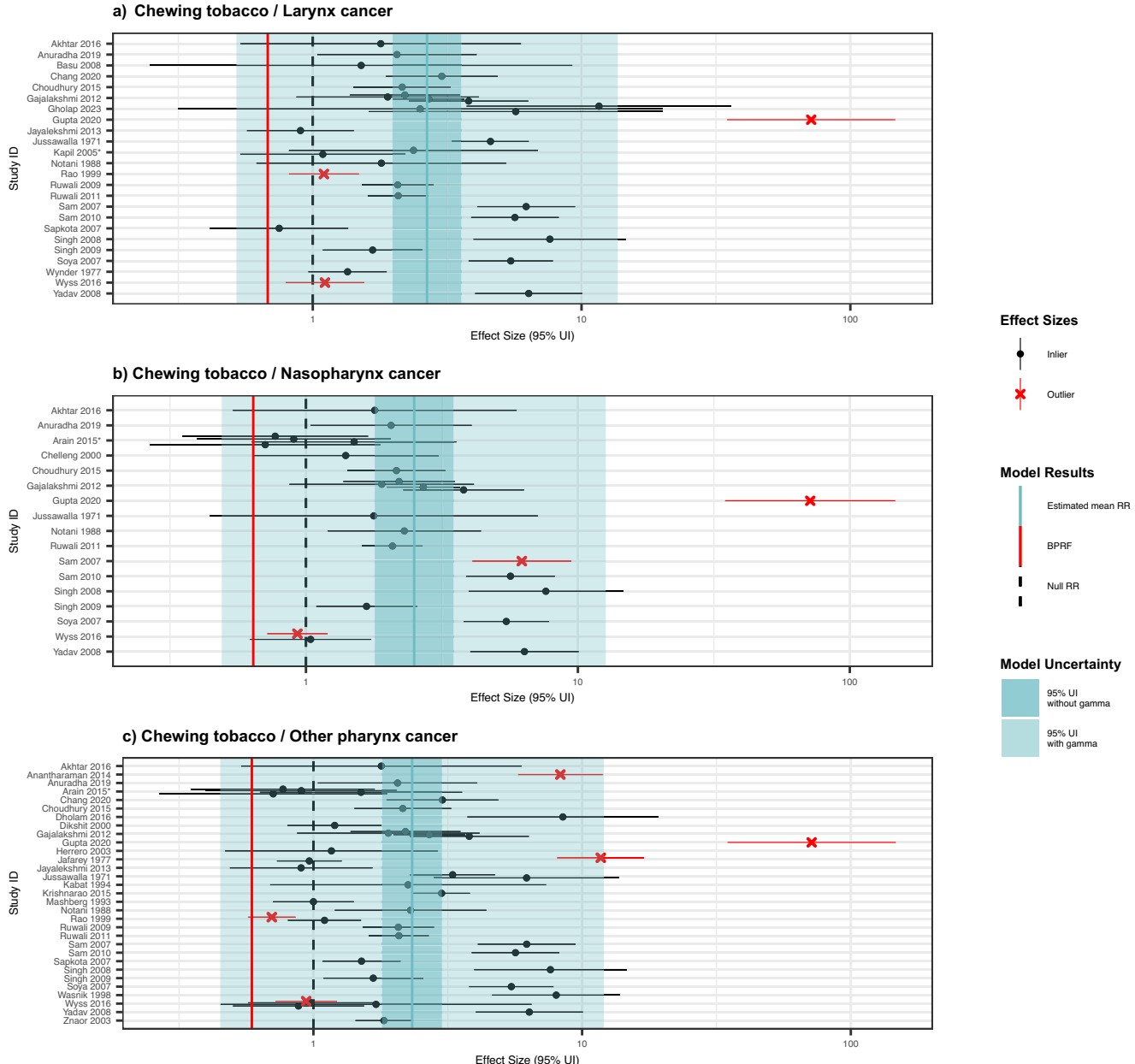

**Fig. 7 | Forest plots of underlying data for chewing tobacco and three other head and neck cancers.** These forest plots depict the estimated mean relative risk (blue vertical line) and its 95% uncertainty interval (blue shaded intervals) for the association between chewing tobacco and laryngeal cancer (panel **a**), for the association between chewing tobacco and nasopharyngeal cancer (panel **b**), and for the association between chewing tobacco and other pharyngeal cancer (panel **c**) and the underlying data points. The narrower darker blue intervals correspond to the 95% uncertainty interval estimated without accounting for between-study heterogeneity in accordance with traditional meta-analytic approaches. The light blue intervals correspond to the 95% uncertainty interval that incorporates between-study heterogeneity and the uncertainty around it. Similarly, the red vertical lines are the Burden of Proof Risk Function (BPRF), which correspond to the

5[th] quantile and is used to derive our risk-outcome score (ROS) for risk-outcome pairs in which the darker blue intervals (the 95% uncertainty interval without between-study heterogeneity) do not include the null value at relative risk = 1. The black dotted vertical lines reflect the null relative risk at 1. The black data points and horizontal lines each correspond to an effect size and 95% uncertainty interval from the study noted in on the y-axes that were included in the models. The red Xs and horizontal lines correspond to effect sizes and 95% uncertainty intervals that were automatically trimmed based on deviation from the means. Studies noted with an asterisk include effect sizes from overlapping samples whose uncertainty interval was scaled based on the number of overlapping observations to avoid over-representing one sample in the models.

tobacco, including chewing tobacco, has a dose-response relationship with some health outcomes, as has been well-documented for smoking[21,103]. Our comparison of health risks among tobacco chewers, regardless of dosage, compared to non-chewers may oversimplify the risk profile associated with chewing tobacco, despite being necessary due to a lack of available data. As more data become available, dose-response risk curves could provide further invaluable insight into how

communities with different use patterns may be affected by chewing tobacco use.

Last, we recognize that the use of data points that employ aggregate outcome definitions for the five head and neck cancer outcomes examined should be regarded with caution. This approach means that we are assuming a given study would report the same effect size for all the outcomes included in its aggregate outcome definition.

However, differences in etiology and location of the cancers means that this assumption is unlikely to be accurate. To account for this limitation, we incorporated the use of a bias covariate to systematically test for differences between outcome-specific effect sizes and the aggregate effect sizes in each of the cancer models, and this covariate was included to adjust aggregate observations in the models where these observations significantly deviated from the rest. We also ran a sensitivity analysis for four of the five head and neck cancer models in which we removed any observation that used an aggregate outcome definition (Supplementary Information 4) and found that one-star risk-outcome associations persisted. The esophageal cancer model evaluating the strength of the evidence was not robust to this change, likely due to the reduced number of data points available and large, persistent between-study heterogeneity, rather than a difference in the overall observed association.

Our study provides a comprehensive examination of the evidence of chewing tobacco's association with each of the seven health outcomes based on data from the literature published in the last 53 years with no restriction to language of publication or location of study. We found that, when accounting for between-study heterogeneity, systematic biases, and other sources of uncertainty, there is weak-to-moderate evidence of an association between chewing tobacco use and stroke and esophageal cancer, highlighting the potential health hazard chewing tobacco presents for the growing number of global users. We affirmed the existence of a large degree of variation in the available literature, which contributes to our finding that there is sufficient, yet weak, evidence of an association between chewing tobacco and laryngeal cancer, lip and oral cavity cancer, nasopharynx cancer, and other pharynx cancer based on a conservative interpretation of the evidence. We further found that there was insufficient evidence of an association between chewing tobacco and ischemic heart disease. Our evaluation of the evidence highlights a need for more high-quality prospective cohort studies evaluating the health impacts of chewing tobacco products among the communities where chewing tobacco is a traditional social norm. In the absence of such research, ambiguity will persist regarding the disease burden attributable to chewing tobacco among communities and countries with high rates of use where chewing tobacco health risks should be an important research priority for informing future tobacco control efforts.

## Methods
### Overview
The present study uses the Burden of Proof Risk Function (BPRF) methodology to produce conservative estimates of the associations between chewing tobacco use and the risks for the health outcomes of interest and to further evaluate the strength of evidence underlying these associations. The BPRF methodology was developed at the Institute for Health Metrics and Evaluation and is summarized in detail by Zheng et al.[18]. The approach has been previously applied to evaluate health risks associated with smoking, high systolic blood pressure, low vegetable consumption, and red meat consumption[18–22]. In brief, the BPRF methodology is a multi-step meta-analytic process: Step 1) Conduct a systematic review of peer-reviewed literature to identify and extract all relevant data on the selected risk-outcome association; Step 2) Estimate a pooled relative risk for the dichotomous risk factor comparing the risk of the outcome for tobacco chewers relative to that of non-tobacco chewers; Step 3) Test whether systematic biases, including those detailed in the Grading of Recommendations Assessment, Development and Evaluation (GRADE) criteria, impact model results and account for any significant biases with included covariates; Step 4) Quantify remaining unexplained between-study heterogeneity ($\gamma$) while accounting for within-study correlations ($\beta$) and incorporate both in the 95% uncertainty estimates; Step 5) Evaluate potential publication bias based on visual examination of funnel plots and Egger's regression test; and Step 6) Generate the BPRF, the 5th quantile

estimate of relative risk closest to the null, and resulting risk-outcome score (ROS). Steps 2 through 6 use MR-BRT (meta-regression−Bayesian, regularized, trimmed), an analytic tool that better accounts for systematic biases in published data, incorporates within- and between-study heterogeneity, and identifies potential outliers than traditional meta-analysis tools[18].

For the purposes of this review, we used the Burden of Proof approach to estimate a separate relative risk, BPRF, and ROS quantifying the relationship between chewing tobacco and each of the seven health outcomes of interest (more details on how the exposure and outcomes were identified and defined are provided below). These models are not location-, sex-, or age-specific but rather draw upon all available data across multiple geographic regions, sexes, and age groups. When sufficient sex-specific data were available, we conducted sex-specific sensitivity analyses that are described in the Supplementary Information alongside other sensitivity analyses described below.

Across the data collection, modeling, and writing, this study adhered to Preferred Reporting Items for Systematic Reviews and Meta-Analyses (PRISMA) guidelines and Guidelines on Accurate and Transparent Health Estimates Reporting (GATHER) recommendations as was feasible (Supplementary Tables S23-S25 and Supplementary Figures S1-S3)[134,135]. This study, as an extension of the Global Burden of Disease study, was approved by the University of Washington Institutional Review Board (study no. 9060). The systematic review protocol was not separately registered but aimed to follow to review best practices.

### Defining chewing tobacco
For the purposes of the present study, chewing tobacco is defined as the current use of chewing tobacco, including local tobacco products that are chewed by the consumer, at any frequency[3]. It is considered distinct from other forms of smokeless tobacco that are not chewed, including snuff, snus, naswar, and gul[4]. Smokeless tobacco products that are chewed and are, thus, encompassed in our definition of chewing tobacco include zarda, gutkha, and mawa[4]. A full list of the chewing tobacco products associated with data points included in our analysis can be found in Supplementary Tables S2 and S4. Notably, betel quid and/or areca nut, which are independently considered carcinogenic, are often associated with chewing tobacco[5]. Because the use of these mixed products is common in many parts of the world with the highest rates of chewing tobacco, our definition of chewing tobacco includes tobacco that is chewed with these additives but omits betel quid or other such substances if they are not mixed with tobacco.

### Selecting the health outcomes of interest
Based on prior research drawing on the World Cancer Research Fund criteria for health associations, esophageal cancer and lip and oral cavity cancer have previously been shown to be associated with chewing tobacco use[17]. However, this study presented an opportunity to apply the BPRF methodology to a broader array of health outcomes. The research team conducted an umbrella review of existing meta-analyses and systematic reviews (MA/SRs) published between January 1, 1970, and December 31, 2021, to identify potential health outcomes that have a demonstrated interest among smokeless tobacco researchers and, therefore, likely have sufficient literature on chewing tobacco specifically to merit a full review. The search string used, and further details of the umbrella review, can be found in the Supplementary Information. The identified MA/SRs were screened by one member of the research team and categorized into their respective outcomes of interest (Supplementary Table S6). The MA/SRs that included outcomes covered by more than one MA/SR underwent a full text review to determine the quality and scope of their included studies. Based on this review and consultation with topic experts, head and neck cancers−namely esophageal cancer, larynx cancer, lip and

 

oral cavity cancer, nasopharyngeal cancer, and other pharynx cancer—and cardiovascular diseases, particularly stroke and ischemic heart disease, were identified as health outcomes of interest with sufficient existing literature to appropriately evaluate the risk-outcome relationship. The definitions of the included outcomes are provided in Supplementary Table S1.

## Step 1) Conducting systematic reviews to identify relevant peer-reviewed literature

We conducted three systematic reviews of all relevant studies indexed in PubMed, Global Index Medicus, and Web of Science between January 1, 1970, and January 30, 2023, as of February 15, 2023. One review was conducted for all head and neck cancers combined since many studies report on different combinations of our five head and neck cancer outcomes. The other two reviews were conducted separately for stroke and for ischemic heart disease. The studies were identified using modified search strings for each database and outcome group (Supplementary Information 1.1), and the hits were uploaded to Covidence, a systematic review management software. Duplicated results between the three databases were automatically flagged by the software and confirmed by a member of the research team.

In brief, studies were excluded if they did not use a cohort or case-control study design, did not report on chewing tobacco or a chewing tobacco product, did not report on the outcome of interest, did not report on an adequate exposure type or exposure category, or focused on a highly specific and nongeneralizable population. Cohort studies and case-control studies were included if they reported an effect size for using chewing tobacco or a chewing tobacco product and the outcome of interest. Using these pre-determined inclusion and exclusion criteria, the titles and abstracts of de-duplicated records were independently screened by two trained team members, and the full text of any record that was included after title/abstract screening underwent further review by two independent screeners. Any conflicting decisions at either title/abstract screening or full text screening were resolved by a third reviewer. Special case protocol applied to MA/SRs and non-English-language sources identified by the search strings, so these were tagged for separate review at the title/abstract screening. Regardless of language, each non-MA/SR source was screened for inclusion by at least two people. MA/SR sources underwent review for potentially relevant underlying citations that were further screened. The search strings for each database, screening protocol for MA/SRs and non-English language sources, and our inclusion and exclusion criteria are described in more detail in the Supplementary Information.

Relevant data were extracted from included studies by a single reviewer using a modified Covidence 2.0 extraction template (Supplemental Table S3). The study metadata characteristics, including location, study design, methods for exposure and outcome ascertainment, and study population demographics, were extracted. The most- and least-adjusted effect sizes (including relative risks, hazard ratios, or odds ratios) reported for chewing tobacco use and the outcomes of interest were also extracted, together with chewing tobacco exposure, reference definition, and temporality (e.g., current chewers compared to former or never chewers; ever chewers compared to never chewers), in addition to the outcome definition used, the corresponding sample sizes, and the covariates adjusted for in each model. If a study did not explicitly report an effect size for chewing tobacco and the outcome but provided enough information to calculate an unadjusted effect size, the extractor would manually calculate the effect size and related uncertainty and extract all the relevant information. Completed extractions were manually reviewed and vetted for accuracy by a second team member. Further information on how the extracted data were cleaned and prepared for modeling can be found in the Supplementary Information.

## Step 2) Estimating the mean association between chewing tobacco and the outcomes of interest

We used the MR-BRT tool to conduct meta-regression analyses with the log-space relative risk of the outcome modeled as the dependent variable and chewing tobacco use as the dichotomous independent variable (current tobacco chewing versus not chewing tobacco). These analyses generated a single pooled relative risk of the association between chewing tobacco and each outcome derived from the extracted effect sizes. To prevent potential outliers from introducing unnecessary noise, we applied 10% trimming to all of the models with more than 10 data points. The process of trimming within MR-BRT applies the least trimmed squares methodology to identify and trim data points that are inconsistent with the patterns reflected in the rest of the dataset based on the number of standard error intervals between the data points and the mean estimate. Models with fewer than 10 data points (i.e., stroke and ischemic heart disease) were ineligible for trimming because of the already limited data availability.

For studies that reported more than one effect size for chewing tobacco and the outcome of interest (e.g., a study that reported the risk of an outcome among both current chewers and ever chewers), we first selected the observations that best matched our outcome and exposure definitions. From these, we marked the most-adjusted effect size for inclusion, prioritizing those that were not sub-group analyses when combined analyses were available. These effect sizes were considered eligible for modeling and were included in our analysis with a minimum of one effect size per included study (See Supplementary Information 2.2 for more details on the criteria used). To account for the fact that some studies reported several effect sizes for multiple non-mutually exclusive exposure groups that passed our selection process (e.g., a single study reporting fully adjusted odds ratios for gutkha users and betel quid with tobacco users for the same outcome without accounting for dual users), we downweighted the standard errors of overlapping data points based on the number of overlapping observations from the study (Supplementary Information 2.2). There are several existing methods for addressing within-study covariance in meta-analyses; however, most require a known within-study covariance matrix or access to participant-level data[136]. In the absence of such information for the present analysis, we have leveraged the limited information we do have available to inform a cautious, yet plausible, adjustment to account for within-study covariance. While this method may yield a very conservative approximation of estimate correlation, it is necessary to appropriately prevent a single study from disproportionately affecting the model results. The effect sizes and standard errors used in our primary analysis are reported in Table S5. To examine the impact of our conservative adjustments, we also ran a sensitivity analysis in which the adjustment factors were not applied, which is described in more detail in Supplementary Information 4.

## Step 3) Testing and adjusting for bias related to variation in study characteristics

We created ten binary covariates to distinguish potential dimensions of systematic bias based on the GRADE approach to bias detection and the unique characteristics of our dataset. These bias covariates account for the representativeness of the study population and analytical sample, methods for ascertaining a participant's exposure and outcome status, deviations in exposure and outcome definitions, and the degree to which potential confounding variables were adjusted for, including smoking. They are described in more detail in Supplementary Information 3.3 together with other potential sources of bias that were considered but not included, and the corresponding bias covariate values for each study are listed in Supplementary Table S9. Covariates were eligible for testing if there were a minimum of two data points in the model with each covariate value. The eligible covariates were tested for inclusion using a covariate selection algorithm, which uses a step-wise Lasso approach to identify bias covariates that

significantly affect the results when included as an interaction term in the primary linear meta-regression. Significant bias covariates were adjusted for in the final mixed-effects model with Gaussian priors intended to avoid overfitting models with small datasets[18].

### Step 4) Quantifying remaining between-study heterogeneity
In addition to the bias covariates, we included a study-level random slope ($\gamma$) to capture remaining between-study heterogeneity and a study-level random intercept for within-study correlation. Because our analysis of some outcomes relied on relatively few input studies, which can contribute to underestimating between-study heterogeneity, we also derived the uncertainty of $\gamma$ using the inverse Fisher Information Matrix to inform draws of $\gamma$. These draws are used to derive the uncertainty estimate for our relative risk with $\gamma$, estimated from both the uncertainty surrounding the mean effect and the 95th quantile of between-study heterogeneity draws. The relative risk without $\gamma$, as reported in Table 2, is reported with an uncertainty derived without fully accounting for between-study heterogeneity and reflects the relative risk estimates that are typically reported in traditional meta-analyses, while that with $\gamma$ better reflects the degree of consistency across the underlying studies.

### Step 5) Evaluating the potential for publication and reporting bias
We used Egger's regression to test for potential publication and reporting bias and confirmed the results by inspecting modified funnel plots visualizing the model results and uncertainty alongside the residual mean and standard deviations of the data.

### Step 6) Estimating the burden of proof risk function (BPRF)
Using our final model for each outcome and chewing tobacco, we estimated the BPRF. The BPRF reflects the most conservative estimate of the harmful association between chewing tobacco use and the outcome that is consistent with the evidence, given the variation between data inputs. It is defined as the 5th quantile of the relative risk estimates closest to null. Using the BPRF, we can quantify the conservative percentage of increased risk associated with chewing tobacco as the (BPRF-1) × 100. From the BPRF, we derived the risk-outcome score (ROS) for dichotomous risk factors as the signed natural log(BPRF) divided by two. The value of the ROS reflects the estimated strength of the association between the risk factor and the outcome, with a large positive ROS indicating that there is a large effect size and strong, consistent evidence of the association between the risk factor and the outcome, a small positive ROS indicating a small effect and inconsistent evidence, and a negative ROS suggesting that there is weak evidence of any significant association. The ROS for a risk-outcome association can be then translated into a star rating ranging from a one-star pair (weak evidence of association with an ROS less than 0.0) to a five-star pair (very strong evidence of an association with an ROS greater than 0.62)[18]. The ROS thresholds for two-, three-, and four-star pairs are 0.0–0.14, >0.14–0.41, and >0.41–0.62, respectively. No star rating or ROS is assigned when the risk-outcome pair is found to have a 95% uncertainty interval estimated without between-study heterogeneity (i.e., using the fixed effects model results prior to the incorporation of gamma) that crosses the null. These risk-outcome pairs are deemed to have insufficient evidence of an association between the exposure and the outcome of interest to evaluate the strength of the evidence, and consequently, do not satisfy the criteria for further evaluation and potential inclusion in the GBD.

### Model validation
MR-BRT has been extensively and rigorously validated by Zheng and colleagues for its use in conducting meta-analyses[18]. For the present study, we conducted several sensitivity analyses to evaluate the robustness of our final results given the limitations of our dataset. These analyses are described in more detail in the Supplementary Information. In brief, for each of the outcomes, except stroke and ischemic heart disease, we ran a sensitivity analysis that omitted trimming 10% of the data. Stroke and ischemic heart disease, which had fewer than 10 included observations, were not eligible for trimming within the primary analysis. For models with significant bias covariates selected in the primary analysis, we examined the impact of running the models without providing any bias covariates for potential inclusion in the model with and without 10% trimming. We also ran an additional sensitivity analysis without downweighting observations derived from nonmutually exclusive analytical samples with and without 10% trimming.

Beyond validating the model parameters, we also ran a number of sensitivity analyses with restrictions on the included data points. For the five cancer outcomes with data points that used aggregate outcome definitions, we tested omitting data points that were not specific to the cancer in question. Observing heterogenous exposure definitions, we also evaluated the impact of omitting observations that used ever-chewing tobacco as the exposure group, keeping only observations that compared current tobacco chewers to nontobacco chewers, and the impact of potential regional differences in chewing tobacco definitions by restricting data to only that from Asian countries, the region with the highest number of chewing tobacco users. Similar heterogeneity was observed in sample sizes, so we evaluated the impact that small sample sizes may be having on model results by restricting the input data to only observations with more than five participants each in the exposed and unexposed case and control groups. Given the potential confounding effect of smoked tobacco among dual users that may not be fully captured in our adjustment bias covariates, we also examined the impact of chewing tobacco derived only using observations from non-smoking study samples. Last, given potential differences in chewing intensity or disease patterns between males and females, we ran two further sensitivity analyses limited to only male-specific observations or female-specific observations, respectively. For these analyses evaluating the robustness of our models to our data input, we kept all other model parameters consistent with our primary analysis and ran them both with and without 10% trimming (for analyses with more than ten observations). These analyses were only possible with three or more eligible observations for a given outcome.

### Reporting summary
Further information on research design is available in the Nature Portfolio Reporting Summary linked to this article.

## Data availability
The findings from this study are supported by data extracted from published literature. The estimates produced in this study have been deposited in the Burden of Proof database, which can be accessed via the Burden of Proof visualization tool (https://vizhub.healthdata.org/burden-of-proof/). The estimates are freely available. The relevant studies were identified through a systematic review of previously published literature and can all be identified online as referenced in the current paper. The processed data from these studies that underlies the estimates are included in the Supplementary Information and can also be found and downloaded from the Burden of Proof visualization tool for use. These data are publicly available for download through the visualization tool with no restrictions to access. Study characteristics and included observations for all input data used in the analyses are also provided in the Supplementary Information, while the resulting estimates can also be found in Table 2, but they can all be found in the Burden of Proof tool described above.

## Code availability

All code used for these analyses is publicly available online ([https://github.com/ihmeuw-msca/burden-of-proof/](https://github.com/ihmeuw-msca/burden-of-proof/))[137]. Analyses were carried out using R version 4.0.5 and Python version 3.10.9.

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

## Acknowledgements

Research reported in this publication was supported by the Bill & Melinda Gates Foundation (Grant number: OOPP1152504; Awardee: CJLM) and Bloomberg Philanthropies (Grant number: 47386; Awardee: EG). The funders of the study had no role in study design, data collection, data analysis, data interpretation, writing of the final report, or the decision to publish.

## Author contributions

S.C., J.W., J.A.A., M.J.M., C.O., and G.F.G. were primarily responsible for seeking, cataloguing, extracting, or cleaning data; designing or coding figures and tables. X.D., K.B., M.J.M., L.S.F., R.J.D.S., E.G., and G.F.G. provided data or critical feedback on data sources. S.I.H., X.D., S.C., K.B., C.J.L.M., L.S.F., R.J.D.S., E.G., and G.F.G provided critical feedback on methods or results. S.I.H., X.D., C.J.L.M., L.S.F., S.A.M., E.G., and G.F.G. drafted the work or revised it critically for important intellectual content. E.M.O., E.M., E.G., and G.F.G. managed the overall research enterprise. X.D., S.C., A.A., P.Z., C.J.L.M., and R.J.DS. developed methods or computational machinery. G.F.G. was primarily responsible for applying analytical methods to produce estimates. G.F.G. wrote the first draft of the manuscript. S.I.H., A.A., E.MO., E.G., and G.F.G. managed the estimation or publication process.

## Competing interests

The authors declare no competing interests.
