## [Peer Review File · Nature Communications]

Health effects associated with chewing tobacco: a Burden of Proof studyREVIEWER COMMENTS

Reviewer #1 (Remarks to the Author):

This is a comprehensive review of the health effects of smokeless tobacco. I have some comments that should be addressed in a revision:

1. In line 52 you refer to "local products". Local to where? the specific region is not stated.
2. In lines 67/68 you cite the 89th IARC monograph (2004) that outlined the association between smokeless tobacco products and cancer. However, there is an updated IARC monograph (Vol 100 E) that evaluated the carcinogenicity of ST among other lifestyle factors.
3. For lip and oral cavity cancers you estimated a relative risk of 3.88. This is the expected result. However, in line 298 you translate this risk to "at least 3-16%". It is not clear how a RR of 3.88 can be as low as a 3-16% increased risk.
4. You did not find sufficient evidence of a significant association between chewing tobacco use and the risk of ischemic heart disease. This finding is contrary to many important epi studies earlier reported from India. Please refer to the high-quality work of P.C. Gupta. Perhaps including some US studies in your analysis (eg ref 26) may have attenuated the risk estimate. Kindly revisit this analysis as this is an important public health message.
5. Under Methods line 445, you have excluded naswar, and gul from the analysis. Naswar, in particular, is a popular ST used in Pakistan and Afghanistan and is considered by many authors as an ST product. Naswar is the major risk factor for oral cancer in this Region. This should be included in the analysis.

Reviewer #2 (Remarks to the Author):

"Health effects associated with chewing tobacco: a Burden of Proof study"

First thoughts on the paper by Gabriela F. Gil et al. submitted to Nature Communications.

Author: P. N. Lee

Date: 21st June 2023

I have conducted numerous meta-analyses related to the effects of smoking and tobacco in

relation to numerous diseases, though I have never looked in detail at the evidence on chewing tobacco. At the present time I have looked at parts of this very large submission, which concludes that there is weak-to-moderate evidence that tobacco chewers have an increased risk of stroke, and of each of the five cancer groupings considered, but there was insufficient evidence of an association with ischaemic heart disease. When I conduct meta-analyses I typically present results of standard fixed-effect meta-analyses and of random-effect meta-analyses, and I looked for these in the material provided. I found Tables S13 to S19 which (inter alia) presented columns headed RR (95% UI without γ) and RR (95% UI with γ), noting that the RRs were always the same, but that the confidence intervals differed. For the primary analysis (results also shown in Table 2 of the main paper) all the without γ RRs (except that for ischaemic heart disease) were significant, but all the with γ RRs (except that for stroke) were not significant, and given the with γ RRs seemed more relevant, it was unclear why the conclusion was reached that there was weak-to-moderate evidence of an association, especially as the with γ RRs were virtually always non-significant in the sensitivity analyses for the five cancer types. I also did not understand why the without and with γ RRs were always the same, as fixed-effect and random-effects RR estimates typically vary.

I decided to do my own meta-analyses for one of the cancers, choosing nasopharynx cancer as it had relatively few estimates. Using the data in Table S5 "Summary of Data Inputs" I then extracted the 24 log effect sizes and standard errors from the 17 studies providing data. One major problem was that for the studies with multiple estimates (Arain 2015 – 4; Gajalakshmi 2012 – 4; and Wyss 2016 – 2) there was no indication whatsoever in Table S5 (or anywhere else in the paper that I could find) to distinguish between the estimates (e.g. were some for males, and some for females, or did they relate to different levels of adjustment for covariates) or say which had been included in the primary analyses which were based on only one estimate from each study. According to my meta-analyses, if one included all 24 estimates, one ended up with a fixed-effect estimate of 2.87 (95% CI 2.57-3.20) and a random-effects estimate of 2.65 (1.84-3.82), and even choosing the largest estimate from the studies with more than one estimate, one still ended up with estimates of fixed – 3.44 (3.05-3.89) and random – 3.42 (2.28-5.14), somewhat different from those given in the paper, with notably the random-effect estimate very clearly significant, consistent with the claims of the paper. I am not at all sure how the authors produce their

with γ RRs – it would seem simpler and better to use random-effects RR estimates.

In the process of doing these calculations, and also looking at the forest plots, various things struck me about study Gupta 2020. First, the RR estimate was enormously high (the log effect size of 4.27 being equivalent to a RR of 71.5, far far greater than that from any of the studies, where the largest other one (Singh 2008) was 7.61. Second, the log effect size estimate for nasopharynx cancer was the same (with the same standard error) for all the 5 cancers studied. Without reference back to the source Singh paper, it seems that that the paper provided an estimate for the five cancers combined, and the authors have taken it to apply to all the cancers individually. This is total nonsense – on a par with saying that if smoking doubles the RR for overall mortality, it also doubles the risk of each individual disease – and anyway the frequency of the 5 types of cancer varies, so clearly the estimates cannot have the same standard error. I would definitely not assume that risk estimates for a broad group of diseases apply to each disease individually, and omit the Gupta 2020 estimate from the analysis of nasopharynx cancer. There are a number of other studies where this error needs correction (e.g. Akhtar 2016, Anuradha 2019, Basu 2008) – if you are presenting results for a disease, one should not include estimates for a broader grouping including the disease.

Looking at the Forest plots, some things struck me. One, why present the studies in reverse alphabetical order (bizarre!). Two, if you are going to include the multiple estimates from a study, can you not indicate which one was included in the primary estimate (or if some combined estimate was produced from the multiples show that). Three, I note that the blue shading indicates the width of the 95% confidence interval of the combined with γ estimate, but as I have noted already these estimates appear to be very much too wide, based on a simple random-effects estimate.

A few other things struck me – though I have not read everything in detail.

In the sensitivity analyses, the results for the primary analysis and for the “no covariates included” are always the same. I can make no sense of this – if the primary analysis includes all the estimates, whether or not covariates were adjusted for, and the “no covariates included” only included unadjusted estimates, it would seem absolutely remarkable that the results could possibly be the same (unless no study adjusted for covariates, which can't be true).

Did any of the studies restrict attention to non-smokers in their analyses of chewing

tobacco? It is generally true that those who use one source of nicotine are more likely to use other sources. Without such restriction, excess risks associated with chewing may be in part due to chewers being more likely to smoke.

What about occupational risks? Chewing is a convenient way of getting nicotine in situations (e.g. down mines) where smoking is not allowed or hazardous, and excess risks for chewers may partly reflect occupational risks.

In general, I thought there was too little information for each RR on what variables had been adjusted for, and which were sex-specific. This could be fitted into Table S5, by using abbreviations for the health outcomes, and much shorter headings for columns 3 and 4. There is some information in Table S9, but this is at the study level, and RRs within study may vary on the extent of adjustment.

REVIEWER COMMENTS

Reviewer #1 (Remarks to the Author):

This is a comprehensive review of the health effects of smokeless tobacco. I have some comments that should be addressed in a revision:

1. In line 52 you refer to "local products". Local to where? the specific region is not stated.

In our original submission, we noted that our definition of chewing tobacco refers to forms of smokeless tobacco that are masticated by the users and used the term "local products" to clarified that this definition also encompasses sub-types of chewing tobacco products that are used in specific communities around the world, like gutkha, mainpuri, and zarda. Per your comment, we removed the term "local" since, as you note, this term inaccurately suggests that there is a specific regional focus in our analysis. Instead, we hope this edit clarifies that our review encompasses any chewing tobacco sub-type/product as long as the primary habitual form of use is mastication.

2. In lines 67/68 you cite the 89th IARC monograph (2004) that outlined the association between smokeless tobacco products and cancer. However, there is an updated IARC monograph (Vol 100 E) that evaluated the carcinogenicity of ST among other lifestyle factors.

Thank you for bringing the updated IARC monograph (Vol 100 E) to our attention! We have reviewed the updated IARC for any studies that our review may have missed to make sure we encompass all of the available evidence on the association between chewing tobacco and the seven health outcomes of interest. Of the case-control or cohort studies mentioned in the updated IARC monograph that were published before 1970, we had already reviewed 36/40 as part of our first submission. We reviewed the remaining four and found that they did not report an effect size for chewing tobacco specifically. Of the studies we had already screened, most similarly examined smokeless tobacco broadly or non-chewed forms of smokeless tobacco, which fall outside of the scope of the present analysis. We confirmed that we have already included all the eligible studies in the updated IARC monograph pertaining to chewed forms of smokeless tobacco. We have also updated the citation and reference in lines 69-71, as well as other locations where the previous IARC monograph was cited, to this more recent update.

3. For lip and oral cavity cancers you estimated a relative risk of 3.88. This is the expected result. However, in line 298 you translate this risk to "at least 3-16%". It is not clear how a RR of 3.88 can be as low as a 3-16% increased risk.

We have added clarification in the Results (lines 147, 192, 231), the Discussion (line 329), in the Methods section (lines 618-620) regarding how we derived the estimated increased risk and made sure to describe this percentage as the minimum estimated increased risk. These percentages refer to the percentage increase in relative risk derived from the conservative interpretation of available evidence used in the burden of proof analytical approach. The relative risk estimate for lip and oral cavity cancers reflects the mean estimate of association, while the Burden of Proof Risk Function (BPRF) and the corresponding average increased risk is the 5th quantile of the relative risk estimates closest to a relative risk of 1. The average increased risk is calculated directly from the (BPRF), rather than from the relative risk point estimate, so it reflects the minimum increased risk of the health outcome experienced by chewing tobacco users based on the most conservative interpretation of all the evidence. 1-16% reflects

the range of increased risk estimated for stroke, lip and oral cavity cancer, and esophageal cancer as the three health outcomes with star ratings of two in the burden of proof framework.

4. You did not find sufficient evidence of a significant association between chewing tobacco use and the risk of ischemic heart disease. This finding is contrary to many important epi studies earlier reported from India. Please refer to the high-quality work of P.C. Gupta. Perhaps including some US studies in your analysis (eg ref 26) may have attenuated the risk estimate. Kindly revisit this analysis as this is an important public health message.

Given the public health importance of ischemic heart disease, we considered it an important health outcome to evaluate and screened over 1500 studies to identify the breadth of available evidence related to the association between chewing tobacco use and the risk of ischemic heart disease. As our analysis is not limited to a specific region to capture all available evidence, we did not exclude studies based on their geography alone. However, as you mentioned and as we described in our Discussion (lines 397-400), there are potential differences in the risk profile of specific products used by some communities, which may translate into regional differences that cannot be fully explored at present time given the very limited amount of data available in most of the world. We modified the text to clarify this point.

Within the constraints of existing literature, we also conducted a new sensitivity analysis for all our outcomes by limiting the studies to only those conducted in Asian countries, including India, Pakistan, and Bangladesh. The results of this sensitivity analysis reflect whether or not the inclusion of studies from countries with fewer users of chewing tobacco (like the US and European countries) may be attenuating our global risk estimates. Specifically for ischemic heart disease, we still did not find sufficient evidence of an association between chewing tobacco use and the risk of ischemic heart disease (See Supplementary Table S14 and Figure 3). This finding remains consistent with other meta-analyses that find heterogeneous results on the relationship between chewing tobacco and ischemic heart disease (Discussion, lines 352-362). In fact, when limiting our data to only Asian studies, our results did not deviate from those of our primary analyses for any of the health outcomes with the exception of esophageal cancer, which saw a lower star rating (going from a two-star risk-outcome pair in our primary analysis to a one-star risk-outcome pair in this sensitivity analysis). We added this sensitivity analysis to our results.

5. Under Methods line 445, you have excluded naswar, and gul from the analysis. Naswar, in particular, is a popular ST used in Pakistan and Afghanistan and is considered by many authors as an ST product. Naswar is the major risk factor for oral cancer in this Region. This should be included in the analysis.

We completely agree with this reviewer's comment that naswar is a popular form of smokeless tobacco used in Pakistan and Afghanistan, however, the present analysis and review focuses solely on chewed forms of smokeless tobacco, distinct from non-chewed forms of smokeless tobacco. Non-chewed and chewed forms of smokeless tobacco can be quite different in terms of their risk profile and composition, in addition to having different forms of administration, which guided our decision to narrow the scope of our review to only chewing tobacco products. As a result, we do not consider naswar, gul, snus, or other such non-chewed forms of smokeless tobacco because their main mode of administration is being placed in the mouth and sucked. This distinction in terms of mode of use is also described in the updated Volume 100E IARC monograph mentioned above¹. While the risk of oral cancer associated with

these forms of non-chewed smokeless tobacco may be significant, evaluating this risk is outside of the scope of this analysis that focuses on the health risks of chewing tobacco.

Reviewer #2 (Remarks to the Author):

“Health effects associated with chewing tobacco: a Burden of Proof study”

First thoughts on the paper by Gabriela F. Gil et al. submitted to Nature Communications.

Author: P. N. Lee

Date: 21st June 2023

I have conducted numerous meta-analyses related to the effects of smoking and tobacco in relation to numerous diseases, though I have never looked in detail at the evidence on chewing tobacco. At the present time I have looked at parts of this very large submission, which concludes that there is weak-to-moderate evidence that tobacco chewers have an increased risk of stroke, and of each of the five cancer groupings considered, but there was insufficient evidence of an association with ischaemic heart disease. When I conduct meta-analyses I typically present results of standard fixed-effect meta-analyses and of random-effect meta-analyses, and I looked for these in the material provided. I found Tables S13 to S19 which (inter alia) presented columns headed RR (95% UI without γ) and RR (95% UI with γ), noting that the RRs were always the same, but that the confidence intervals differed. For the primary analysis (results also shown in Table 2 of the main paper) all the without γ RRs (except that for ischaemic heart disease) were significant, but all the with γ RRs (except that for stroke) were not significant, and given the with γ RRs seemed more relevant, it was unclear why the conclusion was reached that there was weak-to-moderate evidence of an association, especially as the with γ RRs were virtually always non-significant in the sensitivity analyses for the five cancer types. I also did not understand why the without and with γ RRs were always the same, as fixed-effect and random-effects RR estimates typically vary.

We have modified the text of our Methods to clarify the distinction between the RRs with gamma and without gamma. In brief, due to our usage of MR-BRT and the Burden of Proof framework, the difference between the RRs reported with gamma and without gamma is in how we estimate the uncertainty surrounding the mean RR, not in our estimation of the mean RR itself. The modeling methodology is described in detail in the Burden of Proof methods capstone and has been applied and validated in several *Nature Medicine* publications but, as elaborated upon in the Methods of this paper and a Commentary published in *Nature Medicine*, is distinct from traditional fixed and random effects models²⁻⁷.

In traditional meta-analyses, between-study heterogeneity has a limited effect on the posterior uncertainty surrounding the mean effect size, and its impact is attenuated by increasing number of studies, even if these studies contradict each other. The uncertainty interval we report that does not incorporate gamma reflects the uncertainty around the mean effect, akin to what would be reported in a traditional meta-analysis. This estimate reflects the existence of an association between the health outcome and chewing tobacco based on current evidence and is the basis for evaluating whether or not a risk-outcome pair merits inclusion in the Global Burden of Disease study.

With the RR with gamma, we take a step further and additionally incorporate the uncertainty surrounding between-study heterogeneity as well as between-study heterogeneity, which we believe is the appropriate course of action to fully capture the uncertainty within existing literature on the

association between outcome and risk. The final uncertainty intervals we report with gamma are derived from both the posterior uncertainty surrounding the mean effect size and the 95th quantile of between-study heterogeneity, estimated using the inverse of Fisher Information Matrix. This measure, RR with gamma fully incorporated, is reported to reflect the degree to which existing literature consistently finds the association between the health outcome and chewing tobacco. It is not meant to be examined independently of the RR without gamma, but rather provides a complementary measure that captures unexplained sources of variation between studies. The Burden of Proof Risk Function, as the 5th quantile of the RR with gamma, informs our interpretation of the strength of the evidence, again explained in more detail by Zheng et al. (2022) and Aravkin et al. (2023)^{2,7}.

I decided to do my own meta-analyses for one of the cancers, choosing nasopharynx cancer as it had relatively few estimates. Using the data in Table S5 “Summary of Data Inputs” I then extracted the 24 log effect sizes and standard errors from the 17 studies providing data. One major problem was that for the studies with multiple estimates (Arain 2015 – 4; Gajalakshmi 2012 – 4; and Wyss 2016 – 2) there was no indication whatsoever in Table S5 (or anywhere else in the paper that I could find) to distinguish between the estimates (e.g. were some for males, and some for females, or did they relate to different levels of adjustment for covariates) or say which had been included in the primary analyses which were based on only one estimate from each study.

We greatly appreciate the level of detail and time taken to replicate our meta-analysis! We hope others will also be able to draw upon the data provided in Table S5 to validate our estimates. We recognize Table S5 leaves ambiguity as to the differences across observations from the same studies, particularly since they are all used in our primary analyses. We addressed this issue by adding a couple of columns to Table S5, including one that describes the reason why multiple observations from one study may have been selected for inclusion in our primary analyses using our data point selection framework (which is described briefly in the Methods (lines 556-569) and in detail in Supplementary Information Section 2.2). In brief, for studies that reported multiple effect sizes, we selected those that best matched our outcome and exposure definitions, were the most-adjusted reported effect size, and were the most aggregated group if sub-group analyses were available. The selected effect sizes are those reported in Table S5. We modified the Methods (lines 556-569) to clarify this process further. When this selection schema resulted in multiple included observations from a single study based on overlapping samples (for example, two effect sizes reported for two different sub-types of chewing tobacco without accounting for dual users that would be reflected in both), we inflated the corresponding standard error by multiplying it with the square root of the number of included observations from the sample to reduce the over-weighting of a handful of studies in our estimates. These adjusted standard errors are reported in Table S5 as they are the direct data inputs for our primary analysis.

According to my meta-analyses, if one included all 24 estimates, one ended up with a fixed-effect estimate of 2.87 (95% CI 2.57-3.20) and a random-effects estimate of 2.65 (1.84-3.82), and even choosing the largest estimate from the studies with more than one estimate, one still ended up with estimates of fixed – 3.44 (3.05-3.89) and random – 3.42 (2.28-5.14), somewhat different from those given in the paper, with notably the random-effect estimate very clearly significant, consistent with the claims of the paper. I am not at all sure how the authors produce their with γ RRs – it would seem simpler and better to use random-effects RR estimates.

We appreciate the reviewer's approach to validating our relative risk estimates! As described above, all 24 estimates in Table S5 were included in our primary analysis, which reflected adjustments made to account for overlapping study samples. When comparing the reviewer's results with all 24 estimates, we do not find them to be significantly different from our own without the full incorporation of gamma (2.55; 95% UI: 1.83-3.54). The differences in our results can be easily explained by our use of the MR-BRT modeling tool. For models with more than 10 observations, including nasopharynx cancer, we applied 10% trimming to remove interference from outliers, which would not have been used in a simple random-effects or fixed-effects model using all data points. It is also possible that the reviewer did not incorporate the two bias covariates that were detected for inclusion in our own model for nasopharynx cancer.

Our incorporation of gamma in estimating the uncertainty of the RR, which involves both quantifying between-study heterogeneity and uncertainty surrounding between-study heterogeneity, drives the further differences between the reviewer's results and our RRs with gamma, as it does the differences between our RRs with and without gamma. The strengths of this approach have been described above and in a recently published Commentary in *Nature Medicine*.² The Burden of Proof approach presents a comprehensive series of measures, including RRs with gamma, to evaluate the strength of the available evidence. Our use of MR-BRT and how we produce our estimates are described in the Methods (lines 546-609) and the Supplementary Information Section 3.4. We have added more detail on our estimation of the RRs with gamma in the Methods (lines 603-609).

In the process of doing these calculations, and also looking at the forest plots, various things struck me about study Gupta 2020. First, the RR estimate was enormously high (the log effect size of 4.27 being equivalent to a RR of 71.5, far far greater than that from any of the studies, where the largest other one (Singh 2008) was 7.61. Second, the log effect size estimate for nasopharynx cancer was the same (with the same standard error) for all the 5 cancers studied. Without reference back to the source Singh paper, it seems that that the paper provided an estimate for the five cancers combined, and the authors have taken it to apply to all the cancers individually. This is total nonsense – on a par with saying that if smoking doubles the RR for overall mortality, it also doubles the risk of each individual disease – and anyway the frequency of the 5 types of cancer varies, so clearly the estimates cannot have the same standard error. I would definitely not assume that risk estimates for a broad group of diseases apply to each disease individually, and omit the Gupta 2020 estimate from the analysis of nasopharynx cancer. There are a number of other studies where this error needs correction (e.g. Akhtar 2016, Anuradha 2019, Basu 2008) – if you are presenting results for a disease, one should not include estimates for a broader grouping including the disease.

We greatly appreciate the reviewer's attention to detail! We reviewed Gupta 2020 again to confirm the accuracy of our extraction given this very large effect size, and we confirmed that the very high RR estimate was accurately extracted from the reported number of cases and controls for head and neck cancers. In our present models, the trimming algorithm incorporated in the Burden of Proof analysis, described in the Methods (lines 550-555), consistently acts as expected and marks Gupta 2020 as an outlier among the cancer outcomes. Consequently, the high effect size does not affect our relative risk estimates, burden of proof risk function, or uncertainties.

As the reviewer correctly pointed out, for studies that reported on the association between chewing tobacco and a grouping of more than one head and neck cancer, we applied the reported relative risk to

each of the head and neck cancer outcomes included in the given grouping. We believe this is reasonable given that chewing tobacco affects the risk of these different cancers through similar mechanisms and, as the reviewer stated, would not make the same assumption for larger groups of outcomes, like all-cause mortality. Prior to our first submission, we investigated strategies for potentially adjusting the effect sizes or standard errors based on disease incidence to account for the limitations in applying these aggregated outcome definitions, however, we decided against doing so because the majority of studies with aggregate outcome definitions did not report on the proportions of the subtypes included in their sample. Any adjustment made would have introduced even more assumptions, particularly regarding the study populations and sampling strategy, and added additional uncertainty to our estimates that could not be adequately substantiated.

We acknowledge that this is a limitation of our study necessary to leverage the full expanse of existing evidence given the wide range of head and neck cancer groupings that are used in literature. We describe it in our Discussion (lines 408-421) as an important caveat to the interpretation of our results. We also conducted a sensitivity analysis for our cancer outcomes to examine the extent to which the effect sizes derived from aggregate outcome definitions influenced our final results. In this sensitivity analysis, described in the Methods (lines 649-650) and in detail in the Supplementary Information Section 4, we only included studies specific to the given cancer. The results are reported throughout the Results, Figure 3, and Supplementary Information Section 4, and overall, we found this analysis to be largely consistent with our primary analysis.

Looking at the Forest plots, some things struck me. One, why present the studies in reverse alphabetical order (bizarre!).

We have changed the order that the studies are presented on the forest plots in alphabetical order.

Two, if you are going to include the multiple estimates from a study, can you not indicate which one was included in the primary estimate (or if some combined estimate was produced from the multiples show that).

As described in one of our responses above, all the effect sizes reported in Table S5 and in our forest plots are included in the primary analysis. We have modified the captions of the forest plots to clarify this point. We describe our process for selecting these observations briefly in the Methods (lines 556-562) and in more detail in the Supplementary Information Section 2.2. For selected observations from the same study derived from mutually exclusive samples, we did not make any further adjustments. When a study reported more than one observation that aligned with our selection criteria and that were derived from non-mutually exclusive samples, we downweighed its observations by multiplying the standard error by the square root of the number of observations to avoid over-representing a single sample in the results. This process is described in the Methods (lines 562-568).

Three, I note that the blue shading indicates the width of the 95% confidence interval of the combined with γ estimate, but as I have noted already these estimates appear to be very much too wide, based on a simple random-effects estimate.

As described above, the 95th confidence interval with γ reflects was estimated by incorporating both standard estimates of uncertainty as estimated by a random-effects model and the 95th percentile

of between-study heterogeneity. This process is aligned with the Burden of Proof methodology used throughout this analysis.

A few other things struck me – though I have not read everything in detail.

In the sensitivity analyses, the results for the primary analysis and for the “no covariates included” are always the same. I can make no sense of this – if the primary analysis includes all the estimates, whether or not covariates were adjusted for, and the “no covariates included” only included unadjusted estimates, it would seem absolutely remarkable that the results could possibly be the same (unless no study adjusted for covariates, which can’t be true).

We clarified in the Methods (lines 646-648) that the sensitivity analysis that does not include covariates does not refer to a sensitivity analysis only with unadjusted observation, but rather, it involved not evaluating any bias covariates for inclusion in our models. The dataset for this sensitivity analysis was unchanged from the primary analysis. We acknowledge that the results are nearly identical to 2 decimal places between the models that incorporate bias covariates that were found to be significant and those that do not incorporate any bias covariate (the “no covariates included” analysis). The Burden of Proof analytic framework uses strong Gaussian priors in the selection of bias covariates to avoid overfitting the model in the face of limited data for each value of the covariate. More details on the use of the Gaussian priors can be found in the Burden of Proof methods capstone⁷. The use of this prior was also incorporated in the framework to reduce the instability observed in the covariate selection process during its development prior to its application in this study. We have modified the Methods (line 583) and referenced the Methods capstone to highlight the role of the prior as the driver of attenuated differences between the models including covariates and the sensitivity analyses without covariates.

Did any of the studies restrict attention to non-smokers in their analyses of chewing tobacco? It is generally true that those who use one source of nicotine are more likely to use other sources. Without such restriction, excess risks associated with chewing may be in part due to chewers being more likely to smoke.

The reviewer correctly highlights that smoking status is an important consideration when examining the health effects of chewing tobacco. In our original submission, we account for studies that controlled for smoking status or limited their analytical sample by smoking status through our series of dummy covariates that depicted a data point’s degree of adjustment. Observations needed to be controlled for smoking status to be considered maximally adjusted or even appropriately adjusted. These covariates were tested for inclusion in our models through a bias covariate selection algorithm that evaluated whether data points with the given adjustment levels deviated significantly from the data points that were not adjusted for at least smoking status, age, and sex. The covariate flagging observations adjusted for at least age, sex, and smoking status were selected for the primary analysis for esophageal cancer and nasopharynx cancer.

Based on this feedback, we decided to go further and return to the subset of studies that explicitly limited their analysis to only non-smokers. We ran an additional sensitivity analysis by restricting our data to only the observations derived from samples of non-smokers. This limited the number of included observations substantially, relative to our primary analysis, to fewer than 10 for four of our seven outcomes and fewer than the minimum of three necessary to run our meta-analysis for stroke. However, from the analyses we were able to run with and without trimming, we actually did find stronger evidence of an association for other pharynx cancer, larynx cancer, and lip and oral cavity

cancer than in our primary analysis, while our star rating (or lack thereof) remained unchanged for ischemic heart disease and nasopharynx cancer. For esophageal cancer, we had insufficient observations to run our analysis with 10% trimming of outliers, so while the non-smoker sensitivity analysis without trimming resulted in a one-star rating (as opposed to the two-star rating in our primary analysis), this aligned with the untrimmed results from our complete dataset. This evaluation of a subset of our data is a great contribution to the paper, affirms our results as a conservative interpretation of all available data, and bolsters our call for further research attention to chewing tobacco distinct from other forms of tobacco use, including smoking.

What about occupational risks? Chewing is a convenient way of getting nicotine in situations (e.g. down mines) where smoking is not allowed or hazardous, and excess risks for chewers may partly reflect occupational risks.

We appreciate that the reviewer brought up occupational risks in association with chewing tobacco as chewing tobacco is engaged with in occupational settings in some parts of the world. We have added a brief discussion of this question in the Discussion (lines 399-402), however, very few of our included studies adjusted for occupation, so we are limited in our ability to account for this potential source of unmeasured confounding. As you mention, differing occupational risks and occupational profiles of the individuals included in studies conducted in different locations may be one of the factors contributing to the large heterogeneity we observed across studies, so we have made sure to highlight this alongside other potential sources of variation and as a potential area for future meta-analyses to explore when more data stratified by occupation is available.

In general, I thought there was too little information for each RR on what variables had been adjusted for, and which were sex-specific. This could be fitted into Table S5, by using abbreviations for the health outcomes, and much shorter headings for columns 3 and 4. There is some information in Table S9, but this is at the study level, and RRs within study may vary on the extent of adjustment.

Per your suggestion, we've made changes to both Table S5 and S9. As described above, we added three columns to the data-point specific rows in Table S5: 1) A column flagging whether a data point is sex-specific, 2) A column that clarifies the difference between data points for studies with multiple included observations in the primary analysis (eg. one observation for smokers and one for non-smokers), and 3) A column describing, in short form, which bias covariates the data point was flagged for. We believe this level of additional detail for each RR will help make the data point selection process more transparent and improve the replicability of our results with the MR-BRT tool. For Table S9, we clarified that each row corresponds to an observation from the corresponding study.

References:

1. Personal habits and indoor combustions. in *A review of human carcinogens* (ed. Centre international de recherche sur le cancer) vol. 100E (International agency for research on cancer, 2012).
2. Aravkin, A. Y. *et al.* Reply to: Concerns about the Burden of Proof studies. *Nat Med* 1–2 (2023) doi:10.1038/s41591-023-02295-7.

3. Dai, X. *et al.* Health effects associated with smoking: a Burden of Proof study. *Nat Med* **28**, 2045–2055 (2022).
4. Lescinsky, H. *et al.* Health effects associated with consumption of unprocessed red meat: a Burden of Proof study. *Nat Med* **28**, 2075–2082 (2022).
5. Razo, C. *et al.* Effects of elevated systolic blood pressure on ischemic heart disease: a Burden of Proof study. *Nat Med* **28**, 2056–2065 (2022).
6. Stanaway, J. D. *et al.* Health effects associated with vegetable consumption: a Burden of Proof study. *Nat Med* **28**, 2066–2074 (2022).
7. Zheng, P. *et al.* The Burden of Proof studies: assessing the evidence of risk. *Nat Med* **28**, 2038–2044 (2022).

REVIEWER COMMENTS

Reviewer #1 (Remarks to the Author):

Thank you for responding to reviewers' comments

Reviewer #2 (Remarks to the Author):

“Health effects associated with chewing tobacco:

a Burden of Proof study”

Comments on the revised manuscript (NCOMMS-23-21587A)

Author : P.N. Lee

Date : 9th August 2023

I thank the authors for the detailed replies to my rather lengthy comments on the original manuscript. While I myself would have done the meta-analyses differently (using standard fixed-effect and random-effects meta-analyses, and not assuming that relative risks for a combined endpoint apply equally to its component endpoints) the reply convinces me that what has been done seems valid. Partly due to pressure of other work, and partly as it would take me a very considerable time to fully get to grips with the detail of the methodology used (I am nearly 80 and perhaps the saying about teaching old dogs new tricks applies!), I am happy to accept that the revised manuscript is now OK. Certainly the replies to my earlier comments give me confidence in the validity of the revised paper.

Reviewer #3 (Remarks to the Author):

Thank you for the opportunity to review this manuscript, “Health effects associated with chewing tobacco: a Burden of Proof study”. I commend the authors for using a new and complex method for analysis that holds great promise for the field of meta-analysis and for writing a clear and easily interpreted manuscript. My expertise is in statistical methods for

meta-analysis, and as such my critiques will focus on those aspects of this manuscript.

1. Greater justification is needed for the authors' choice of how they inflated standard errors of effect size estimates from the same study by multiplying by the square root of shared sample size. While I agree some adjustment must be made for correlated effects (as is common in multivariate meta-analyses), the authors' approach to this appears overly punitive and introduces unnecessary noise in their analysis, potentially by an order of magnitude (see example below). I highly suggest the authors either incorporate multivariate meta-analysis methods and estimate estimation error covariances (which can be done from 2x2 tables), or find a more reasonable approach to adjusting for this correlation.

a. As an example, consider two effect size estimates from the same study that are perfectly correlated (i.e., they use identical participants and their outcomes; basically the same effect size used twice) computed on 100 total observations (shared between both effects) and denote their common estimation error variance as v . If both effects were incorporated into a fixed-effects meta-analysis (i.e., if you include an effect twice), then that one effect (which is included twice) would receive twice the weight it should; the weight given to this one effect (which is included twice) would be $2/v$. The solution, then, would be to multiply the standard errors by $\sqrt{2}$, which would effectively halve the weight each of the effects received (i.e., both effects would receive a weight proportional to $1/2v$, and hence overall the one effect that is repeated would get weight proportional to $1/v$, rather than $2/v$). However, under the authors' approach, one would multiply the standard errors by 10 in this example, which would give each effect a weight proportional to $1/100v$ and would give the overall effect a weight proportional to $1/50v$. Further, since standard errors in meta-analyses are inverse functions of the weights, not only does this approach give too little weight to the study in this example, it also inflates uncertainty in estimated parameters to an unnecessary degree.

2. Zheng et al. argue for trimming outliers in the context of estimating exposure-risk curves via spline models. However, given that does not appear to be a step in the present analysis, why do the authors opt for outlier exclusion in their primary analysis? I would think that

their inclusion of all studies in their “sensitivity analysis” would be more reflective of the data. If my understanding of the methods used in this manuscript is correct (see below), the models presented are ostensibly mixed effects meta-regressions, and seldom are outliers trimmed in the fitting of those models (without the identification of scientific or analytic flaws in the studies excluded). As well, some of the trimmed effects displayed in forest plots do not visually appear to be striking outliers.

3. I appreciate that this is a complex estimation procedure in general and that describing it step-by-step can be useful to the non-statistical readers, but as a statistician, I had a hard time figuring out exactly what the final model was. My understanding is that you ultimately fit a meta-regression model with random study effects and included variables selected from a stepwise LASSO procedure. Is this correct?

4. In the context of the present manuscript, the exposure is binary and hence the spline portion of the methods described by Zheng et al. appear not to be included in the present analysis, is this correct?

5. The step-by-step description (both here and in the article by Zheng et al.) also makes it a little unclear if fixed and random effects in the final models are estimated jointly or if the method uses a plug-in estimate for the variance component.

6. I have some confusion regarding how the BPRF is calculated: are relevant quantiles determined from a full a posterior predictive distribution of effect parameters? Or is there a different type of calculation using the conservative 95th percentile of the estimated between-study variance component? If it is the latter, I suggest reporting a similar quantity using the posterior predictive distribution of effect parameters as that reflects a more conventional Bayesian statement regarding updated beliefs about the impact of chewing tobacco given the data.

7. The inclusion of a random study effect in the final model allows for heterogeneity between effects in different studies, as well as a correlation between effects in the same study. However, I wonder if the authors considered within-study variation as an important source of effect heterogeneity, and if so, the methods outlined in the present manuscript and by Zheng et al. do not appear to include parameters for that type of variation.

REVIEWER COMMENTS

Reviewer #1 (Remarks to the Author):

Thank you for responding to reviewers' comments

We are grateful for the opportunity to respond to original comments provided by the reviewers and for the time taken to review our updated manuscript.

Reviewer #2 (Remarks to the Author):

“Health effects associated with chewing tobacco: a Burden of Proof study”

Comments on the revised manuscript (NCOMMS-23-21587A)

Author : P.N. Lee

Date : 9th August 2023

I thank the authors for the detailed replies to my rather lengthy comments on the original manuscript. While I myself would have done the meta-analyses differently (using standard fixed-effect and random-effects meta-analyses, and not assuming that relative risks for a combined endpoint apply equally to its component endpoints) the reply convinces me that what has been done seems valid. Partly due to pressure of other work, and partly as it would take me a very considerable time to fully get to grips with the detail of the methodology used (I am nearly 80 and perhaps the saying about teaching old dogs new tricks applies!), I am happy to accept that the revised manuscript is now OK. Certainly the replies to my earlier comments give me confidence in the validity of the revised paper.

We appreciated the attention to detail in the original comments by this reviewer because they gave us the opportunity to strengthen our submission and clarify some of the more complex components of the burden of proof methodology. We are heartened to hear that our responses thoroughly addressed the reviewer’s concerns and appreciate the reviewer’s decision.

Reviewer #3 (Remarks to the Author):

Thank you for the opportunity to review this manuscript, “Health effects associated with chewing tobacco: a Burden of Proof study”. I commend the authors for using a new and complex method for analysis that holds great promise for the field of meta-analysis and for writing a clear and easily interpreted manuscript. My expertise is in statistical methods for meta-analysis, and as such my critiques will focus on those aspects of this manuscript.

Given that this manuscript reflects the first application of the burden of proof methodology to chewing tobacco, we appreciate the additional care taken to review the statistical strengths of the novel approach. We thank the reviewer for their kind words and hope we responded satisfactorily to their questions below.

1. Greater justification is needed for the authors’ choice of how they inflated standard errors of effect size estimates from the same study by multiplying by the square root of shared sample size. While I agree some adjustment must be made for correlated effects (as is common in multivariate meta-analyses), the authors’ approach to this appears overly punitive and introduces unnecessary noise in their analysis, potentially by an order of magnitude (see

example below). I highly suggest the authors either incorporate multivariate meta-analysis methods and estimate estimation error covariances (which can be done from 2x2 tables), or find a more reasonable approach to adjusting for this correlation.

- a. As an example, consider two effect size estimates from the same study that are perfectly correlated (i.e., they use identical participants and their outcomes; basically the same effect size used twice) computed on 100 total observations (shared between both effects) and denote their common estimation error variance as v . If both effects were incorporated into a fixed-effects meta-analysis (i.e., if you include an effect twice), then that one effect (which is included twice) would receive twice the weight it should; the weight given to this one effect (which is included twice) would be $2/v$. The solution, then, would be to multiply the standard errors by $\sqrt{2}$, which would effectively halve the weight each of the effects received (i.e., both effects would receive a weight proportional to $1/2v$, and hence overall the one effect that is repeated would get weight proportional to $1/v$, rather than $2/v$). However, under the authors' approach, one would multiply the standard errors by 10 in this example, which would give each effect a weight proportional to $1/100v$ and would give the overall effect a weight proportional to $1/50v$. Further, since standard errors in meta-analyses are inverse functions of the weights, not only does this approach give too little weight to the study in this example, it also inflates uncertainty in estimated parameters to an unnecessary degree

We believe there may have been some miscommunication regarding our process for inflating the standard errors, and we have updated the main manuscript text (lines 547/548 and figure captions) to prevent such miscommunication in the future. We do not inflate the standard error by leveraging the shared sample size because, as the reviewer highlights, this would result in a dramatic under-weighting of the effect sizes. Rather, we utilize the number of included effect sizes from the same study with overlapping samples. In this, we align with the solution proposed by the reviewer of multiplying the standard errors of two effect estimates from the same study sample by the square root of two. In our submitted manuscript, we referred to this process as multiplying the standard errors by the square root of the number of observations, with observations referring to the included effect sizes, but this language is imprecise, so we have updated the text to refer to number of included effect sizes to prevent further confusion.

2. Zheng et al. argue for trimming outliers in the context of estimating exposure-risk curves via spline models. However, given that does not appear to be a step in the present analysis, why do the authors opt for outlier exclusion in their primary analysis? I would think that their inclusion of all studies in their "sensitivity analysis" would be more reflective of the data. If my understanding of the methods used in this manuscript is correct (see below), the models presented are ostensibly mixed effects meta-regressions, and seldom are outliers trimmed in the fitting of those models (without the identification of scientific or analytic flaws in the studies excluded). As well, some of the trimmed effects displayed in forest plots do not visually appear to be striking outliers.

While the focus of Zheng et al. was on the application of the burden of proof analysis to continuous risk exposures, the rationale for incorporating trimming in models with sufficient data points is consistent in our dichotomous models. Trimming with MR-BRT, our analytical tool, is an approach that has been

presented in prior burden of proof studies for both continuous and dichotomous risks by detecting observations that substantially deviate from the patterns reflected in the data given their reported uncertainty. In testing the removal of such data points with simulated data, Zheng et al. found that algorithmic likelihood-based trimming improved the stability and reliability of resulting risk estimates. Trimming outliers is particularly important for Burden of Proof analyses given that our results speak to the degree of consistency in published literature, which would be quite affected by the inclusion of data points that substantially deviate from an otherwise consistent field. As with the rest of our methodology, we aligned our approach with the guidance provided by Zheng et al. However, for the purpose of transparency, we also ran all our models with more than 10 observations with and without trimming and described these in both the main text (Figure 3) and the Supplementary Information.

3. I appreciate that this is a complex estimation procedure in general and that describing it step-by-step can be useful to the non-statistical readers, but as a statistician, I had a hard time figuring out exactly what the final model was. My understanding is that you ultimately fit a meta-regression model with random study effects and included variables selected from a stepwise LASSO procedure. Is this correct?

Yes, this is exactly right. The final model is a linear mixed effects model with random effects by study (with gamma denoting their variance), that uses bias covariates that encode study design and were selected by the LASSO stepwise procedure. Our focus while writing the manuscript was to ensure that all readers, particularly policy makers who may find our results actionable, are able to follow, however, we appreciate that having a clear statement pointing to what the final model is in technical terms is also very useful for fellow researchers. As such, we have clarified this point by adding a sentence akin to our explanation in this response in the Methods (lines 437-439).

4. In the context of the present manuscript, the exposure is binary and hence the spline portion of the methods described by Zheng et al. appear not to be included in the present analysis, is this correct?

Yes. While the methods proposed by Zheng et al. can be (and have been) applied to binary and continuous exposures, the exposure definition used in the present manuscript is binary. As a result, the spline portion of the methodology is not relevant nor described in our main text because it is only used to better inform the shape of continuous dose-response relationships.

5. The step-by-step description (both here and in the article by Zheng et al.) also makes it a little unclear if fixed and random effects in the final models are estimated jointly or if the method uses a plug-in estimate for the variance component.

The fixed and random effects in the final linear mixed effects model are estimated jointly. We have added clarification to this regard in the Methods (lines 440-441).

6. I have some confusion regarding how the BPRF is calculated: are relevant quantiles determined from a full a posterior predictive distribution of effect parameters? Or is there a different type of calculation using the conservative 95th percentile of the estimated between-study variance component? If it is the latter, I suggest reporting a similar quantity using the posterior predictive distribution of effect parameters as that reflects a more conventional Bayesian statement

regarding updated beliefs about the impact of chewing tobacco given the data.

The Burden of Proof Risk Function (BPRF) is a new quantity introduced by the Burden of Proof methodology that is calculated as described in the manuscript (lines 581-584) and in Zheng et al. As a result, BPRF is a measure that incorporates both between-study heterogeneity and its uncertainty obtained using the Fisher Information Matrix, rather than a standard Bayesian sampling approach. This aspect of the methodology makes it robust to cases with low study numbers, like a handful of the risk-outcome pairs described in the present manuscript, as shown in simulations described by Zheng et al. As reported in previous published Burden of Proof analyses for other risk factors, the BPRF is interpreted in conjunction with other measures reported in Table 2 to reflect our comprehensive understanding of the impact of chewing tobacco on these health risks and the current breadth of available literature.

7. The inclusion of a random study effect in the final model allows for heterogeneity between effects in different studies, as well as a correlation between effects in the same study. However, I wonder if the authors considered within-study variation as an important source of effect heterogeneity, and if so, the methods outlined in the present manuscript and by Zheng et al. do not appear to include parameters for that type of variation.

Since there are only a handful of studies with multiple effect sizes reported, we do not consider it as an important source of heterogeneity for the present analysis and do not include parameters for this variation. Instead, we go beyond traditional approaches to meta-analyses by fully incorporating estimates of between-study heterogeneity and the uncertainty of said estimates. Getting at within-study variation is more challenging since authors typically report results for the entire study, particularly for dichotomous risk-outcome pairs. With increasing coordination between research groups, it may become possible to address this additional challenge and look forward to doing so in future research efforts.

REVIEWER COMMENTS

Reviewer #3 (Remarks to the Author):

I would like to commend the authors on their prompt and clearly written responses to my previous review. Now that I understand the authors' methods a little better, I have one remaining critique and one suggestion.

1. Though the authors clarified that their adjustment to estimation error variances (or standard errors) of effect sizes from the same study pertains to the number of effect sizes in that study (as opposed to the number of individual participants), it still needs better justification. The hypothetical I provided in my previous review would be considered an extreme adjustment; multiplying the standard errors of two effect estimates from the same study (which provides only two effect estimates) would be appropriate if those estimates were perfectly correlated. In this sense, the authors are almost certainly overinflating estimation errors (or equivalently, they are appropriately inflating them if they assume estimation errors are perfectly correlated within studies, which is testable though likely untrue).

As a simplified example, assume there is no between-study variance ($\gamma = 0$), and that n effect estimates in the same study have estimation error correlation ρ and the same estimation error variance (v). Then the weight each would receive should be $1/((n-1) * \rho * v + v)$. Note that this simplifies to the authors' proposed solution only if $\rho = 1$. However, if the estimation correlation is smaller, say, 0.3, then for a study with $n=2$ effects, the authors' approach would assign a weight of $1/2v$ for each effect, but the optimal approach would be to assign $1/1.3v$ weight to each (i.e., the authors' approach would assign only 65% of the optimal weight to these effects). Two points are worth noting about this. First, the question of weights gets more complicated in the presence of between-study heterogeneity ($\gamma > 0$). Second, the impact of using this extreme model for estimation correlations is that their resulting parameter estimates are potentially much noisier than they need to be.

In a frequentist meta-analysis, robust variance estimation procedures, including sandwich estimators, can adjust analyses (specifically the uncertainty in estimated parameters) for the fact that estimation error covariances are typically not reported and thus unknown to the analyst. Because the authors use a Bayesian approach, there does not appear to be a

standard analog for these methods. However, there are some options that remain open, even in a Bayesian paradigm. First, the authors could present the analysis as-is but caveat that it represents an extreme model for estimation errors unlikely to be true, but that is likely (potentially very) conservative in terms of uncertainty. Alternatively, they could use their fitted model to impute estimation covariances using principles outlined by Wei and Higgins. Finally, they could opt to specify various multivariate models for estimation error (i.e., varying the correlations from 0 to 1) in their Bayesian meta-regression and fit their models assuming those correlations were known.

2. I understand and (mostly) accept the authors' argument regarding trimming. I would note that trimming in this context has two implications. First, excluding (or substantially down-weighting) the results of specific studies for parameter estimation because they are not consistent with other studies will on its face raise validity concerns with consumers of meta-analyses, particularly if they are high-quality studies. Second, there are several potential models in the meta-analytic literature that have been used to explain why results may vary and adjust estimates accordingly (in some cases, quite efficiently), including a large literature on publication selection. I note this because while I find the methods they used promising and refreshing, this will remain a point of concern for methodologists and applied researchers alike as they figure out how the models used in BOP meta-analyses map onto some well-studied phenomena about published research.

Reference:

Wei, Y. and Higgins, J.P. (2013), Estimating within-study covariances in multivariate meta-analysis with multiple outcomes. *Statist. Med.*, 32: 1191-1205.

<https://doi.org/10.1002/sim.5679>

REVIEWER COMMENTS

Reviewer #3 (Remarks to the Author):

I would like to commend the authors on their prompt and clearly written responses to my previous review. Now that I understand the authors' methods a little better, I have one remaining critique and one suggestion.

We appreciate the reviewer's continued focus on our methodology and are heartened to hear that our previous responses helped clarify our work further. We hope our response provided below and resulting revisions will appropriately address your remaining feedback.

1. Though the authors clarified that their adjustment to estimation error variances (or standard errors) of effect sizes from the same study pertains to the number of effect sizes in that study (as opposed to the number of individual participants), it still needs better justification. The hypothetical I provided in my previous review would be considered an extreme adjustment; multiplying the standard errors of two effect estimates from the same study (which provides only two effect estimates) would be appropriate if those estimates were perfectly correlated. In this sense, the authors are almost certainly overinflating estimation errors (or equivalently, they are appropriately inflating them if they assume estimation errors are perfectly correlated within studies, which is testable though likely untrue).

As a simplified example, assume there is no between-study variance ($\gamma = 0$), and that n effect estimates in the same study have estimation error correlation ρ and the same estimation error variance (v). Then the weight each would receive should be $1/((n-1) * \rho * v + v)$. Note that this simplifies to the authors' proposed solution only if $\rho = 1$. However, if the estimation correlation is smaller, say, 0.3, then for a study with $n=2$ effects, the authors' approach would assign a weight of $1/2v$ for each effect, but the optimal approach would be to assign $1/1.3v$ weight to each (i.e., the authors' approach would assign only 65% of the optimal weight to these effects). Two points are worth noting about this. First, the question of weights gets more complicated in the presence of between-study heterogeneity ($\gamma > 0$). Second, the impact of using this extreme model for estimation correlations is that their resulting parameter estimates are potentially much noisier than they need to be.

In a frequentist meta-analysis, robust variance estimation procedures, including sandwich estimators, can adjust analyses (specifically the uncertainty in estimated parameters) for the fact that estimation error covariances are typically not reported and thus unknown to the analyst. Because the authors use a Bayesian approach, there does not appear to be a standard analog for these methods. However, there are some options that remain open, even in a Bayesian paradigm. First, the authors could present the analysis as-is but caveat that it represents an extreme model for estimation errors unlikely to be true, but that is likely (potentially very) conservative in terms of uncertainty. Alternatively, they could use their fitted model to impute estimation covariances using principles outlined by Wei and Higgins. Finally, they could opt to specify various multivariate models for estimation error (i.e., varying the correlations from 0 to 1) in their Bayesian meta-regression and fit their models assuming those correlations were known.

While we understand and agree with the reviewer that our approach reflects a conservative approach to accounting for effect sizes derived from non-mutually exclusive study samples, we arrived at the present method upon reviewing the available data and finding that our included studies typically do not report

the proportion of sample overlap (e.g. dual users of two different chewing tobacco sub-types) between included observations as a measure of estimate correlation. Without the consistent availability of such information and given the constraints of our modeling approach, we opted for a cautious approach in which we apply a conservative adjustment derived from the available data. It is worth noting that this adjustment is only applied to a handful of observations included in our models, so we do not expect the impact of this adjustment to be substantial on our overall results.

In alignment with the reviewer's first suggestion, we have further clarified the justification for our standard error inflation in our main manuscript (lines 546-548) and added an explicit acknowledgement that, although the limitations of the data make our approximation of estimation correlation necessary, this approach results in a very conservative adjustment (lines 548-849).

2. I understand and (mostly) accept the authors' argument regarding trimming. I would note that trimming in this context has two implications. First, excluding (or substantially down-weighting) the results of specific studies for parameter estimation because they are not consistent with other studies will on its face raise validity concerns with consumers of meta-analyses, particularly if they are high-quality studies. Second, there are several potential models in the meta-analytic literature that have been used to explain why results may vary and adjust estimates accordingly (in some cases, quite efficiently), including a large literature on publication selection. I note this because while I find the methods they used promising and refreshing, this will remain a point of concern for methodologists and applied researchers alike as they figure out how the models used in BOP meta-analyses map onto some well-studied phenomena about published research.

We appreciate the reviewer flagging the novelty of our trimming methodology and are heartened that they agree in our stance that it presents a promising approach for systematically detecting outliers. This approach has been previously applied in several published and accepted Burden of Proof analyses, and it was derived from trimming methodology used in other types of statistical analyses.¹⁻⁵ In its development, it was further tested in different data sparseness scenarios, and it was found to not affect the quality of the resulting estimates while decreasing the risk of publication or reporting bias.¹ Given that it is likely one component of our analysis that will, as you mention, potentially raise further discussion, we also took the step of running our analyses with and without 10% trimming as additional sensitivity analyses (Figure 3; Supplementary Information Section 4), and we hope this degree of transparency with regard to the impact of trimming will aide in conceptualizing our findings within the context of other meta-analytic approaches.

Response references:

1. Dai, X. et al. Health effects associated with smoking: a Burden of Proof study. *Nat Med* 28, 2045–2055 (2022).
2. Lescinsky, H. et al. Health effects associated with consumption of unprocessed red meat: a Burden of Proof study. *Nat Med* 28, 2075–2082 (2022).
3. Razo, C. et al. Effects of elevated systolic blood pressure on ischemic heart disease: a Burden of Proof study. *Nat Med* 28, 2056–2065 (2022).

4. Stanaway, J. D. et al. Health effects associated with vegetable consumption: a Burden of Proof study. *Nat Med* 28, 2066–2074 (2022).
5. Zheng, P. et al. The Burden of Proof studies: assessing the evidence of risk. *Nat Med* 28, 2038–2044 (2022).

Reference:

Wei, Y. and Higgins, J.P. (2013), Estimating within-study covariances in multivariate meta-analysis with multiple outcomes. *Statist. Med.*, 32: 1191-1205. <https://doi.org/10.1002/sim.5679>.

REVIEWER COMMENTS

Reviewer #3 (Remarks to the Author):

Thank you for the opportunity to review this manuscript. I see that the authors have placed the caveat that analyses are conservative since their approach to dependent effect sizes is conservative. This statement could be made more precise by conducting sensitivity analyses (e.g., making no adjustment to SEs, or assuming/imputing certain values for estimation correlations and fitting the resulting models) or by presenting information on the number of effect sizes and the impact of their adjustments on those effect size weights in analyses.

More generally, while I appreciate the BOP methodology, I question the authors' adherence to it in the face of model misspecification. Clearly the covariance structure in their final model is not properly specified. The authors adjustment to standard errors, though conservative, is hardly necessary as there is a sizable literature on meta-analytic methods for estimation of meta-regression models in the face of unknown covariance structure, including Bayesian methods discussed in the article cited in my previous review.

REVIEWER COMMENTS

Reviewer #3 (Remarks to the Author):

Thank you for the opportunity to review this manuscript. I see that the authors have placed the caveat that analyses are conservative since their approach to dependent effect sizes is conservative. This statement could be made more precise by conducting sensitivity analyses (e.g., making no adjustment to SEs, or assuming/imputing certain values for estimation correlations and fitting the resulting models) or by presenting information on the number of effect sizes and the impact of their adjustments on those effect size weights in analyses.

We appreciate the reviewer's continued focus on helping us improve the precision of our manuscript and strengthen our methodology. We have implemented both of the reviewer's suggestions in the revised version of our manuscript. Specifically, we added the adjustment factors to the Supplementary Table S5 listing all the included effect sizes, and we also conducted a sensitivity analysis in which we do not adjust the standard errors of observations from overlapping samples. Furthermore, we re-evaluated all the adjustment factors to make sure they were appropriately applied and removed a handful of factors where their use was not clearly indicated due to unambiguously overlapping samples. We have updated our findings correspondingly (Main text lines 161-298) and included the findings of our new sensitivity analysis in an updated Figure 3 and the Supplementary Information Section 4 alongside the other sensitivity analyses conducted. Removing the standard error adjustment did not result in any substantial changes to the mean relative risk estimated across our outcomes of interest; the largest absolute change in the mean relative risk was for lip and oral cavity cancer which increased from 3.64 (95% UI without between-study heterogeneity: 3.00-4.41; 95% UI with between-study heterogeneity: 0.66-19.95) in our primary analysis to 3.73 (95% UI without between-study heterogeneity: 3.08-4.53; 95% UI with between-study heterogeneity: 0.66-21.08) in the new sensitivity analysis (Figure 3; Supplementary Information Section 4). Similarly, the star ratings did not change for laryngeal cancer, nasopharyngeal cancer, other pharyngeal cancer, lip and oral cavity cancer, and stroke. Ischemic heart disease was persistently found to have insufficient evidence of an association with chewing tobacco use. The one change we did find in star ratings was for esophageal cancer. Esophageal cancer was categorized as a two-star risk-outcome pair in our primary analysis and a one-star risk-outcome pair in the new sensitivity analysis. This finding aligns with our expectation, given that it had a risk-outcome score (ROS) near the threshold between one- and two-star risk-outcome pairs and removing the standard error adjustment increased the observed between-study heterogeneity that is incorporated in our ROS estimates.

More generally, while I appreciate the BOP methodology, I question the authors' adherence to it in the face of model misspecification. Clearly the covariance structure in their final model is not properly specified. The authors adjustment to standard errors, though conservative, is hardly necessary as there is a sizable literature on meta-analytic methods for estimation of meta-regression models in the face of unknown covariance structure, including Bayesian methods discussed in the article cited in my previous review.

We take the reviewer's comment about model specification as concern that there are correlations between outcomes that are unknown and unaccounted for. We appreciate the concern and have

carefully reviewed and cited the reviewer's suggested reference as a means to address the issue when the covariance between outcomes is in fact reported. However, we respectfully point out that potential correlation structure will be an issue to any meta-analysis – essentially every model that makes any or no assumptions on the correlation structure would be mis-specified. The key point is that the methods provided in the paper referenced by the reviewer are not directly applicable here, since within-study correlations between outcomes were not reported. We do not have access to within-study correlations of related outcomes that could defensibly be used nor do we have access to the individual participant data. We respectfully but firmly disagree with the reviewer that a Bayesian technique, where priors are used with no observed data, is an appropriate method to use in our case – given that the studies we use in our analyses do not report between-outcome correlations, posteriors for within-study between-outcome observations will be equal to the priors, meaning that we would be dictating the between-outcome correlations.

The challenges articulated here and highlighted by the reviewer are not unique to chewing tobacco but well exemplified by it. Many risk factors for health are not well-studied, resulting in meta-analyses with studies that leverage small or unreported sample sizes and unknown covariance structures. Considering this constraint, even the authors of the cited article state that using a plausible value for the unknown covariance matrix is an acceptable means of addressing this challenge. Our approach leverages a conservative, yet plausible, approach that can be used consistently across risk-outcome pairs in Burden of Proof analyses including and beyond the seven described in the present manuscript. We believe using this cautious technique is preferable in the absence of more information. We acknowledge that there are ways to improve the Burden of Proof methodology when within-study correlations are known or can be reasonably inferred, as described in Wei and Higgins, and we hope to make these methods available within the Burden of Proof framework during future development and applications. We have elaborated on this briefly in the limitation section of our Discussion (lines 379-382).

While our adjustment technique represents our approach to address the issue, we ran the sensitivity analysis recommended by the reviewer, and described in more detail above, and found that the adjustment did not have a substantial impact on our final results. We have also extended the section of our Methods describing the standard error adjustment to highlight the rationale for our chosen approach, potential alternatives, and our new sensitivity analysis (lines 549-555). We thank the reviewer for giving us the opportunity to continue to improve our present manuscript as well as highlight important directions of future methodological development for the Burden of Proof analytic approach as it is applied to different use cases.

Reference:

Wei, Y. and Higgins, J.P. (2013), Estimating within-study covariances in multivariate meta-analysis with multiple outcomes. *Statist. Med.*, 32: 1191-1205. <https://doi.org/10.1002/sim.5679ts>.

REVIEWERS' COMMENTS

Reviewer #3 (Remarks to the Author):

I appreciate the authors' sensitivity analysis, which provides important insight into the robustness of their findings to the specification of within-study correlations. I recommend publication of the article and look forward to their future work tackling within-study correlation in BOP methodology.

REVIEWERS' COMMENTS

Reviewer #3 (Remarks to the Author):

I appreciate the authors' sensitivity analysis, which provides important insight into the robustness of their findings to the specification of within-study correlations. I recommend publication of the article and look forward to their future work tackling within-study correlation in BOP methodology.

We are thankful for the reviewer's suggestion to conduct the sensitivity analysis because we agree that it strengthens our submission and provides important further context to our findings. We greatly appreciate the reviewer's decision and the level of attention the reviewer provided to our methodology.